# Suitability Filter: A Statistical Framework
# for Classifier Evaluation in Real-World Deployment Settings

**Angéline Pouget** [1]  **Mohammad Yaghini** [2]  **Stephan Rabanser** [2]  **Nicolas Papernot** [2]

## Abstract

Deploying machine learning models in safety-critical domains poses a key challenge: ensuring reliable model performance on downstream user data without access to ground truth labels for direct validation. We propose the *suitability filter*, a novel framework designed to detect performance deterioration by utilizing *suitability signals*—model output features that are sensitive to covariate shifts and indicative of potential prediction errors. The suitability filter evaluates whether classifier accuracy on unlabeled user data shows significant degradation compared to the accuracy measured on the labeled test dataset. Specifically, it ensures that this degradation does not exceed a pre-specified margin, which represents the maximum acceptable drop in accuracy. To achieve reliable performance evaluation, we aggregate suitability signals for both test and user data and compare these empirical distributions using statistical hypothesis testing, thus providing insights into decision uncertainty. Our modular method adapts to various models and domains. Empirical evaluations across different classification tasks demonstrate that the suitability filter reliably detects performance deviations due to covariate shift. This enables proactive mitigation of potential failures in high-stakes applications.

## 1. Introduction

Machine learning (ML) models often operate in dynamic, uncertain environments. After a model is tested on a holdout set, a satisfactory evaluation result typically leads to production deployment. However, if test and deployment covariate

---

[1]ETH Zurich, work performed while interning at the University of Toronto and the Vector Institute [2]University of Toronto and Vector Institute. Correspondence to: Angéline Pouget <angeline.pouget@gmail.com>.

*Proceedings of the $42^{nd}$ International Conference on Machine Learning*, Vancouver, Canada. PMLR 267, 2025. Copyright 2025 by the author(s).

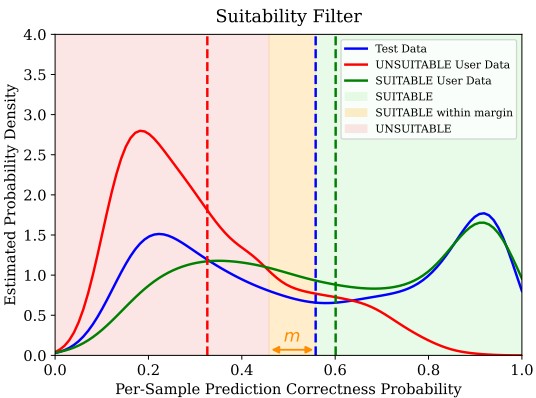

*Figure 1.* **A model** $M$ **is** *suitable* **for use on** $D_u$ **if its accuracy does not fall below the accuracy on** $D_{\text{test}}$ **by more than a predefined margin** $m$**.** The suitability filter calculates per-sample prediction correctness probabilities for both test and user datasets and compares the two distributions through statistical non-inferiority testing. The dashed vertical lines represent the mean values of the distributions corresponding to the estimated accuracies.

distributions differ, performance can drop and cause harm. For example, credit risk models trained on limited historical data may fail in new contexts, disproportionately harming underserved communities through unfair denials or higher interest rates (Kozodoi et al., 2022). Ideally, deployed predictions could be compared directly to ground truth for real-time performance monitoring. However, ground truth may be unavailable (e.g., limited expert labeling (Culverhouse et al., 2003)), unobservable (e.g., counterfactual outcomes in healthcare (Tal, 2023)), or only available much later (e.g., recidivism in law enforcement (Travaini et al., 2022)), thereby causing significant monitoring challenges in deployment.

In this work, we tackle the challenge of determining whether classification accuracy on *unlabeled* user data degrades significantly compared to a labeled holdout dataset—an issue not directly addressed by existing methods. Our approach combines insights from distribution shift detection, unsupervised accuracy estimation, selective prediction, and dataset inference into a novel performance deterioration detector.

Central to our solution is the *suitability filter*, an auxil-

iary function $f_s : \mathcal{X} \to \{\text{SUITABLE, INCONCLUSIVE}\}$. Given an unlabeled user dataset $D_u \sim \mathcal{D}_{\text{target}}$ sampled from the target deployment domain, and a labeled test dataset $D_{\text{test}} \sim \mathcal{D}_{\text{source}}$ sampled from the original training domain, the filter assesses whether classifier accuracy on $D_u$ falls below that on $D_{\text{test}}$ by more than a predefined margin $m$. Our work proposes both (i) a framework for the suitability filter; as well as (ii) a well-performing default instantiation of the filter using domain-agnostic *suitability signals* that are broadly applicable across classifiers, independent of the model architecture or the training algorithm.

To arrive at its decision, the suitability filter relies on suitability signals (model output features such as maximum logit/softmax or predictive entropy). These signals are sensitive to covariate shifts and can indicate potential prediction errors. In particular, we design a per-sample prediction correctness estimator leveraging these suitability signals. This allows us to assess consistency in the model's predictive behavior on both $D_u$ and $D_{\text{test}}$ by aggregating sample-level suitability signals. As a result, we are able to detect subtle shifts indicative of changes in model performance. As illustrated in Figure 1, we then compare the means of these distributions (i.e., the estimated accuracies) to arrive at a suitability decision. Our decisions rely on statistical testing to assess whether the estimated difference in means is significant, thus offering a measure of predictive uncertainty.

To ensure the reliability of suitability decisions, we study the statistical guarantees for the suitability filter. Specifically, we identify the theoretical conditions that ensure a bounded false positive rate for end-to-end suitability decisions. We also consider the practical scenarios where such condition may not hold and provide a relaxation of this theoretical condition. This allows model providers to ensure reliability of suitability decisions in spite of theoretical limitations.

Building on these theoretical insights, we empirically show that the filter consistently detects performance deviations arising from various covariate shifts, including temporal, geographical, and subpopulation shifts. Specifically, we assess the effectiveness of our approach using real-world datasets from the WILDS benchmark (Koh et al., 2021). These include FMoW-WILDS for land use classification (Christie et al., 2018), CivilComments-WILDS for text toxicity classification (Borkan et al., 2019), and RxRx1-WILDS for genetic perturbation classification (Taylor et al., 2019). Furthermore, we explore how accuracy differences between user and test datasets impact the filter's sensitivity, analyze calibration techniques to control false positives, and conduct ablations on suitability signals, sample sizes, margins, significance levels, and classifier options.

In summary, our key contributions are the following:

1. We introduce suitability filters as a principled way of de-

tecting model performance deterioration during deployment. Our filters detect covariate shift via an unlabeled representative dataset provided by the model user.

2. We propose a statistical testing framework to build suitability filters that aggregate various signals and output a suitability decision. Leveraging formal hypothesis testing, our approach enables control of the false positive rate via a user-interpretable significance level.

3. We theoretically analyze the end-to-end false positive rates of our suitability filters and provide sufficient conditions for bounded false positive rates. We then consider a practical relaxation of this condition, and suggest an adjustment to the prediction margin that maintains our end-to-end bounded false error guarantees.

4. We demonstrate the practical applicability of suitability filters across $29k$ experiments on realistic data shift scenarios from the WILDS benchmark. On FMoW-WILDS, for example, we are able to detect performance deterioration of more than $3\%$ with $100\%$ accuracy as can be seen in Figure 4.

## 2. Related Work

Our work builds on insights from distribution shifts, accuracy estimation, selective prediction, and dataset inference.

**Distribution Shift Detection.** Distribution shift detection methods aim to identify changes between training and deployment data distributions (Quiñonero-Candela et al., 2022), generally requiring access to ground truth labels. Early research emphasizes detecting shifts in high-dimensional data using approaches such as statistical testing on model confidence distributions (Rabanser et al., 2019) or leveraging model ensembles (Ovadia et al., 2019; Arpit et al., 2022). Recent efforts increasingly prioritize interpreting shifts (Kulinski & Inouye, 2023; Koh et al., 2021; Gulrajani & Lopez-Paz, 2021) and mitigating their impact on model performance (Cha et al., 2022; Zhou et al., 2021; Wiles et al., 2021; Zhou et al., 2022; Wang et al., 2022). Some works argue that while small shifts are unavoidable, the focus should be on harmful shifts that lead to significant performance degradation (Podkopaev & Ramdas, 2021; Ginsberg et al., 2022). These approaches aim to detect covariate shifts and subsequently assess their impact on performance. To do so, they rely on ground truth labels or model ensembles to evaluate harmfulness. This assumption makes these techniques unsuitable for our setting where we aim to detect performance degradation without label access.

**Unsupervised Accuracy Estimation.** Unsupervised accuracy estimation, also known as AutoEval (Automatic

Model Evaluation (Deng & Zheng, 2021)), aims to estimate a model's classification accuracy (a continuous metric) on unseen data without relying on ground truth labels. Early approaches in this field primarily centered on model confidence, calculated as the maximum value of the softmax output applied to the classifier's logits, and related metrics which we demonstrate to be valuable suitability signals (Hendrycks & Gimpel, 2016; Garg et al., 2022; Kivimäki et al., 2024; Bialek et al., 2024; Guillory et al., 2021; Lu et al., 2023; Wang et al., 2023; Hendrycks & Dietterich, 2018; Deng et al., 2023). Our work differs from these approaches in three key ways: we focus on *reliably* detecting performance *deterioration* (a binary decision) *in relation* to a labeled test dataset using statistical testing.

**Selective Classification.** Selective classification techniques aim to detect and reject inputs a model would likely misclassify, while maintaining high coverage and accepting as many samples as possible (Chow, 1957; El-Yaniv et al., 2010). In contrast to selective classification, we do not reject or accept individual input data samples. Instead, we leverage sample-level signals and aggregate them to provide a statistically grounded suitability decision for the entire dataset. Initial selective classification methods for neural networks base the rejection mechanism on the model prediction confidence (Hendrycks & Gimpel, 2016; Geifman & El-Yaniv, 2017), a signal that we also leverage in our work.

**Dataset Inference.** Our approach is inspired by dataset inference (Maini et al., 2021), a technique used to determine whether a model was trained on a particular dataset. Similarly to dataset inference, we compare suitability distributions between two different data samples through statistical hypothesis testing. However, in contrast to dataset inference, we focus only on evaluation and aim to detect possible performance deterioration, essentially reversing the null and alternative hypotheses. Moreover, dataset inference relies on representative data from both sample domains—the original source and the deployed target domain—to train a confidence regressor. Instead, we assume that label access is only available in data sampled from the source domain.

## 3. Problem Formulation

Our suitability filter framework distinguishes between the *model provider*, who trains and tests the classifier on the source distribution, and the *model user*, who applies the model to (possibly distributionally shifted) target data.

**Model Provider.** Let $\mathcal{Y} = \{1, \ldots, k\}$ denote the label space, representing the set of all possible output labels for a classification problem with $k$ classes. We define our predictor as a model $M : \mathcal{X} \to \mathcal{Y}$ mapping inputs from a covariate space $\mathcal{X}$ to classification decisions. A model provider trains

such a model $M$ on *labeled* data sampled from a source distribution $\mathcal{D}_{\text{source}}$ over domain $\mathcal{X}$. Specifically, the provider usually partitions the data into two disjoint subsets: a training dataset $D_{\text{train}} \sim \mathcal{D}_{\text{source}}$, which is used to optimize the parameters of $M$ and a test dataset $D_{\text{test}} \sim \mathcal{D}_{\text{source}}$, which is reserved to evaluate the performance of $M$ on unseen data. To ensure an unbiased evaluation of model performance, these two datasets are disjoint, i.e., $D_{\text{train}} \cap D_{\text{test}} = \emptyset$.

**Model User.** A model user interested in deploying model $M$ on their data provides an *unlabeled*, representative data sample $D_u \sim \mathcal{D}_{\text{target}}$. In most scenarios of practical interest, $\mathcal{D}_{\text{target}}$ differs from $\mathcal{D}_{\text{source}}$, i.e., $\mathcal{D}_{\text{target}} \neq \mathcal{D}_{\text{source}}$. Note that if $D_u$ were to be drawn from the same distribution as $D_{\text{test}}$, the model's performance on both datasets would be identical in expectation, eliminating the need for the suitability filter. The user might be a third party looking to use the model, or the model provider and user could be the same party.

**Suitability Filter.** The suitability filter assesses whether the performance of classifier $M$ on unlabeled user data $D_u$ degrades relative to the known performance on the labeled test dataset $D_{\text{test}}$. In our work, we focus on model accuracy as the performance metric. We define suitability as follows:

**Definition 3.1** (Suitability). Given a classifier $M : \mathcal{X} \to \mathcal{Y}$, a test data sample $D_{\text{test}} \sim \mathcal{D}_{\text{source}}$, a user data sample $D_u \sim \mathcal{D}_{\text{target}}$, and a performance deviation margin $m \in \mathbb{R}$, we define $M$ as suitable for use on $D_u$ if and only if the estimated accuracy of $M$ on $D_u$ deviates at most by $m$ from the accuracy on $D_{\text{test}}$. Formally:

$$\frac{1}{|D_u|} \sum_{x \in D_u} \mathbb{I}\{M(x) = \mathcal{O}(x)\} \geq$$
$$\frac{1}{|D_{\text{test}}|} \sum_{(x,y) \in D_{\text{test}}} \mathbb{I}\{M(x) = y\} - m. \quad (1)$$

Here, $\mathbb{I}\{\cdot\}$ is the indicator function and $\mathcal{O}(x)$ represents an oracle that provides the true label $y$ for any input $x$ (the ground truth label is unavailable for samples $x \in D_u$).

**Definition 3.2** (Suitability Filter). Given a model $M$, a test dataset $D_{\text{test}}$, a user dataset $D_u$, a performance metric $g$ and a performance margin $m$ as in Definition 3.1, we define a suitability filter to be a function $f_s : \mathcal{X} \to \{\text{SUITABLE}, \text{INCONCLUSIVE}\}$ that outputs SUITABLE if and only if $M$ is suitable for use on $D_u$ according to Definition 3.1 with high probability and INCONCLUSIVE otherwise.

## 4. Method

The suitability filter is introduced as a statistical hypothesis test designed to assess if the performance of a model on user data $D_u$ deviates from its performance on a test dataset $D_{\text{test}}$ by more than a specified margin $m$. By aggregating

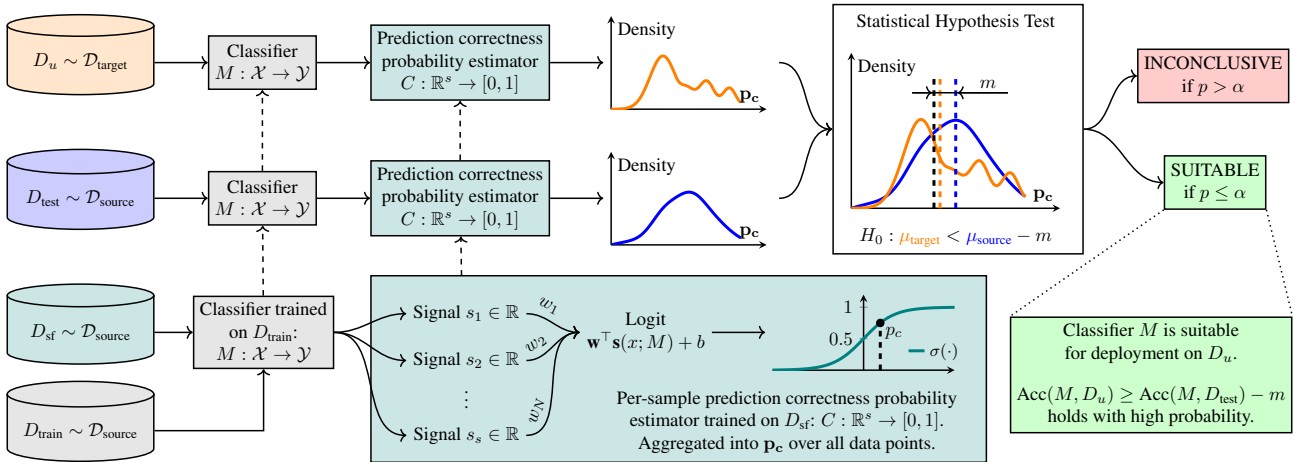

*Figure 2.* **Schematic overview of the suitability filter**. The suitability filter assesses whether model performance on a user sample $D_u$ deviates from its performance on the test dataset $D_{\text{test}}$. This is achieved by combining different suitability signals $\{s_1, \ldots, s_s\}$ to estimate per-sample prediction correctness and comparing the distribution of these estimates between the two datasets using a statistical test.

a diverse set of suitability signals predictive of classifier correctness, the test compares predicted accuracy between $D_u$ and $D_{\text{test}}$ using a non-inferiority test to ensure the mean performance difference does not exceed the performance margin $m$ (Wellek, 2002; Walker & Nowacki, 2011). We present an schematic overview of our approach in Figure 2.

### 4.1. Suitability Signals

The first step in constructing the suitability filter is to select a set of signals $\{s_1, \ldots, s_S\}$ that are predictive of per-sample classifier prediction correctness. These signals are inherently dependent on the model $M$ and capture information about its predictions and confidence levels. As discussed in Section 2, a variety of signals have been proposed in the literature on unsupervised accuracy estimation, selective classification, and uncertainty quantification. Such signals include but are not limited to the maximum logit/softmax scores, the energy of the logits, or the predictive entropy. The exact signals used in this work have been selected to ensure the broad applicability of the suitability filter across diverse settings as outlined in more detail in our experiments (Section 5) and in Appendix A.4.2 and A.4.3. We note that any signal that can be computed for an individual sample and is predictive of prediction correctness can be incorporated into our framework, allowing for flexible extension based on the specific task, dataset, or model $M$.

### 4.2. Per-Sample Prediction Correctness Estimator

To learn a per-sample prediction correctness estimator, we require the model provider to have a separate, labeled hold-out dataset $D_{\text{sf}} \sim \mathcal{D}_{\text{source}}$. While ultimately the goal is to assess performance on the unlabeled $D_u \sim \mathcal{D}_{\text{target}}$ provided by the user, the hold-out dataset $D_{\text{sf}}$ serves as a proxy to

train the parameters of the prediction correctness estimator. This dataset is essential because it enables the suitability filter to learn the relationship between different signals and classifier prediction correctness. $D_{\text{sf}}$ has to be separate from both $D_{\text{train}}$ and $D_{\text{test}}$ to avoid overfitting to these samples.

For each sample $x \in D_{\text{sf}}$, the selected signals $\{s_1, \ldots, s_S\}$, which are functions of both the sample and the model $M$, are evaluated, normalized, and aggregated into a single feature vector $\mathbf{s}(x; M) = [s_1(x; M), s_2(x; M), \ldots, s_S(x; M)] \in \mathbb{R}^S$. The suitability filter framework leverages this feature vector $\mathbf{s}(x; M)$ to predict whether the model $M$ correctly classifies the input $x$. This is achieved by training a prediction correctness classifier $C : \mathbb{R}^S \rightarrow \{0, 1\}$ that estimates the per-sample probability of prediction correctness $p_c(x)$ on the hold-out dataset $D_{\text{sf}}$. In particular, we want to minimize the binary cross-entropy loss between the true correctness label $c = \mathbb{I}\{M(x) = y\}$ and $p_c(x)$ for each $(x, y) \in D_{\text{sf}}$. We instantiate $C$ as a logistic regressor[1] which models the prediction correctness probability $p_c(x) = \sigma(\mathbf{w}^\top \mathbf{s}(x; M) + b)$, where $\sigma(z) = \frac{1}{1+e^{-z}}$ is the sigmoid function.

We can then leverage $C$ to estimate per-sample prediction correctness for user data samples $x \in D_u$ (since calculating $p_c(x)$ does not require ground truth label access) as well as the test data $D_{\text{test}}$. Next, we discuss steps to verify and ensure that $C$ generalizes effectively to $D_u \sim \mathcal{D}_{\text{target}}$.

**Calibration.** To ensure that the mean estimated probability of prediction correctness directly reflects accuracy, $p_c(x)$

---

[1]Note that while more flexible model classes can be used for the correctness estimator $C$, we did not find any empirical evidence that they provide a consistent performance improvement over logistic regression (see Appendix A.4.4 for details).

must be well-calibrated for both samples from $\mathcal{D}_{\text{source}}$ and $\mathcal{D}_{\text{target}}$. However, absent specific assumptions about the differences between $\mathcal{D}_{\text{source}}$ and $\mathcal{D}_{\text{target}}$, achieving this desired calibration is impossible in practice (David et al., 2010).

One reasonable assumption under which such calibration issues can be mitigated is if the potential target distributions consist of subpopulations of the source distribution. In credit scoring, for instance, the target distribution may include subpopulations $\mathcal{S}$, such as minority groups or individuals with limited credit histories, who are underrepresented in the training data. In such scenarios, multicalibration techniques can ensure that $C$ provides accurate predictions for every subpopulation $\mathcal{S} \in \mathcal{C}$, thereby improving reliability across all possible $\mathcal{D}_{\text{target}}$ (Hébert-Johnson et al., 2018). Here, $\mathcal{C}$ denotes a collection of computationally identifiable subsets of the support of $\mathcal{D}_{\text{source}}$ and $i \sim \mathcal{S}$ is a sample drawn from $\mathcal{D}_{\text{source}}$ conditioned on membership in $\mathcal{S}$.

When no assumptions about $\mathcal{D}_{\text{source}}$ and $\mathcal{D}_{\text{target}}$ can be made, achieving reliable calibration is challenging. Calibrating $C$ on $D_{\text{sf}}$ (e.g., using Platt's method (Platt et al., 1999) or temperature scaling (Guo et al., 2017)) ensures that the classification correctness estimator $C$ provides reliable estimates of the probability that model $M$ correctly classifies samples in $D_{\text{sf}} \sim \mathcal{D}_{\text{source}}$. While, in theory, this calibration extends to $D_{\text{test}} \sim \mathcal{D}_{\text{source}}$, we generally cannot assume calibration on $D_u \sim \mathcal{D}_{\text{target}}$. Our approach to addressing this issue combines ongoing quality assurance checks with appropriate margin adjustments and will be discussed in more detail in Section 4.4.

## 4.3. Non-Inferiority Testing

Non-inferiority testing is a statistical method used to assess whether the performance of a new treatment, model, or method is not significantly worse than a reference or control by more than a pre-specified margin $m$ (Wellek, 2002; Walker & Nowacki, 2011). Unlike other statistical tests, which typically test for a difference between distributions, this test aims to confirm that the new method is not inferior by more than a margin of $m$. Consequently, the null hypothesis is that the method is inferior, in contrast to the usual null hypothesis of no difference.

**Correctness Distributions.** If the per-sample prediction correctness estimator $C$ is well-calibrated, the mean of the estimated prediction correctness probabilities across a dataset approximates the accuracy of the model $M$ on that dataset. Formally, let $\mathbf{p_c}[D_{\text{test}}]$ and $\mathbf{p_c}[D_u]$ denote the vectors of estimated prediction correctness probabilities for the test dataset $D_{\text{test}}$ and the user dataset $D_u$, respectively:

$$\mathbf{p_c}[D_{\text{test}}] := \left[p_c(x_1), \ldots, p_c(x_{|D_{\text{test}}|})\right] \in [0,1]^{|D_{\text{test}}|} \quad (2)$$

$$\mathbf{p_c}[D_u] := \left[p_c(x_1), \ldots, p_c(x_{|D_u|})\right] \in [0,1]^{|D_u|} \quad (3)$$

Here, $p_c(x_i)$ represents the estimated probability of prediction correctness for each sample $x_i$.

**Hypothesis Setup.** We define the true means of the estimated prediction correctness probabilities for data drawn from the source and target distributions as follows:

$$\mu_{\text{source}} := \mathbb{E}_{x \sim \mathcal{D}_{\text{source}}}\left[p_c(x)\right] \quad (4)$$

$$\mu_{\text{target}} := \mathbb{E}_{x \sim \mathcal{D}_{\text{target}}}\left[p_c(x)\right] \quad (5)$$

The primary goal of the non-inferiority test is to compare the true mean predicted correctness between the two distributions and determine whether $\mu_{\text{target}}$ is not lower than $\mu_{\text{source}}$ by more than a pre-specified margin $m$. This is formally expressed as the following hypothesis testing setup:

$$H_0 : \mu_{\text{target}} < \mu_{\text{source}} - m \quad (6)$$

$$H_1 : \mu_{\text{target}} \geq \mu_{\text{source}} - m \quad (7)$$

The null hypothesis $H_0$ posits that the estimated performance on the user dataset is worse than on the test dataset by more than the margin $m$. The alternative hypothesis $H_1$ asserts that the estimated performance on the user dataset is either better than, equivalent to or not worse than that on the test dataset within the allowed margin $m$. We conduct the statistical non-inferiority test using a one-sided Welch's t-test (see Appendix A.1.1).

## 4.4. Suitability Decision

Finally, the decision on the suitability of the model for the user dataset is based on the outcome of this non-inferiority test. If the test indicates non-inferiority, we conclude that the model's performance on $D_u$ is acceptable and we output *SUITABLE*. If the test fails to reject the null hypothesis, the model is either unsuitable for the user dataset or the number of samples provided was insufficient to determine suitability and hence we return *INCONCLUSIVE*. To ensure the reliability of these suitability decisions, we next discuss statistical guarantees and the conditions under which they hold for the end-to-end suitability decision.

**Statistical Guarantees.** To account for miscalibration errors, we define $\delta$-calibration as follows:

**Definition 4.1** ($\delta$-Calibration). Let $p_c(x)$ denote the estimated probability of prediction correctness for a sample $x$ with predicted label $M(x)$ and true label $y$. Assuming that $p_c(x)$ has a well-defined probability density function $f_c(\nu)$ over $[0,1]$, we say $p_c(x)$ is $\delta$-calibrated if

$$\mathbb{P}\left[M(x) = y \mid p_c(x) = \nu\right] = \nu + \epsilon(\nu), \quad (8)$$

$\forall \nu \in [0,1]$ with calibration error $\int_0^1 \epsilon(\nu) f_c(\nu)\, d\nu = \delta$ for $0 \leq |\delta| \ll 1$.

Under the assumption of testing two independent and normally distributed samples, the non-inferiority test ensures a controlled false positive rate (FPR), bounding the probability of incorrectly concluding non-inferiority.

**Theorem 4.2** (Non-Inferiority Test Guarantee). *Let $\mu_{source}$ and $\mu_{target}$ represent the true mean prediction correctness for the source and target distributions, respectively. Assuming that these samples are independent and normally distributed, a non-inferiority test based on Welch's t-test at significance level $\alpha$ guarantees that the probability of rejecting the null hypothesis $H_0 : \mu_{target} < \mu_{source} - m$ (i.e., concluding $\mu_{target} \geq \mu_{source} - m$) when $H_0$ is true is controlled at $\alpha$:*

$$\mathbb{P}(Reject\ H_0 \mid H_0\ is\ true) \leq \alpha, \qquad (9)$$

*where $m$ is the non-inferiority margin (Lehmann et al., 1986; Wellek, 2002).*

The following results extend this guarantee to the end-to-end suitability filter under $\delta$-calibration for the correctness estimator $C$ with respect to both $\mathcal{D}_{source}$ and $\mathcal{D}_{target}$. All expectations and probabilities are over samples $(x, y) \sim \mathcal{X} \times \mathcal{Y}$ unless specified otherwise.

**Lemma 4.3** (Expectation of Correctness). *Under $\delta$-calibration as defined in Definition 4.1, the deviation between the true probability of prediction correctness and the estimation by classifer $C$ is given by:*

$$\mathbb{E}[p_c(x)] - \mathbb{P}[M(x) = y] = \delta. \qquad (10)$$

*Proof in Appendix A.1.2.*

We use Lemma 4.3 to derive the end-to-end guarantee for the suitability filter.

**Corollary 4.4** (Bounded False Positive Rate for Suitability Filter under $\delta$-Calibration). *Given a prediction correctness estimator $C$ that is $\delta$-calibrated on both the source and target distributions with $\delta_{source}$ and $\delta_{target}$, respectively, let us define $m' := m + \delta_{source} - \delta_{target}$ and conduct a non-inferiority test with $H_0 : \mu_{target} < \mu_{source} - m'$. The probability of incorrectly rejecting $H_0$ (i.e., returning SUITABLE) when the model accuracy on $\mathcal{D}_{target}$ is lower than on $\mathcal{D}_{source}$ by more than a margin $m$ is upper bounded by the significance level $\alpha$. Proof in Appendix A.1.3.*

The following remark details the limits of these guarantees. *Remark* 4.5 (Impossibility of Bounded False Positive Rate without $\delta$-Calibration). If the calibration deviations $\delta_{source}$ and $\delta_{target}$ are not provided or are not much smaller than 1, it is not possible to choose $m'$ according to Corollary 4.4. Hence, without $\delta$-calibration, no guarantees on the false positive rate of the suitability filter can be provided.

**Practical Considerations.** Under perfect calibration, the calibration errors vanish, i.e., $\delta_{source} = \delta_{target} = 0$, eliminating the need for any margin correction. However, achieving

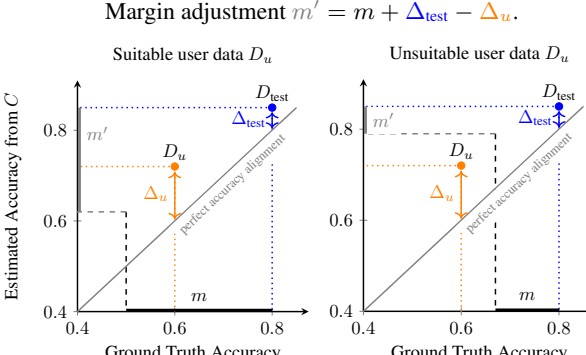

Margin adjustment $m' = m + \Delta_{test} - \Delta_u$.

*Figure 3.* **Margin adjustment under accuracy estimation error**. In each panel, the solid gray line is the perfect-calibration diagonal, the dashed black/gray lines mark the original margin $m$ and its corrected value $m'$, respectively. The blue/orange arrows indicate the estimation errors on the test set ($\Delta_{test}$) and user data ($\Delta_u$), respectively. In the left panel, the user data $D_u$ is deemed suitable; in the right panel it is deemed unsuitable.

perfect calibration in practice is rare. In most real-world deployments, accurately determining the calibration errors $\delta_{source}$ and especially $\delta_{target}$ can be difficult. Consequently, adjusting the margin as proposed in Corollary 4.4 may be challenging. To address this, we draw inspiration from best practices in quality assurance and propose that the model owner periodically collects a small labeled dataset, $\hat{D}_u$, from a potential user of the system. Given access to the test dataset $D_{test}$, the model owner can compute both the estimated accuracies ($\mu_u$, $\mu_{test}$) as approximated by $C$, as well as ground truth accuracies ($Acc_u$, $Acc_{test}$). This enables an empirical evaluation of accuracy estimation errors, $\Delta_u$ and $\Delta_{test}$, which correspond to $\delta_{target}$ and $\delta_{source}$, respectively:

$$\Delta = \frac{1}{N} \sum_{i=1}^{N} p_c(x_i) - \mathbb{I}\{M(x_i) = y_i\} \qquad (11)$$

Following the margin adjustment in Corollary 4.4, the updated margin is:

$$m' = m + \Delta_{test} - \Delta_u. \qquad (12)$$

The intuition behind this adjustment is that the decisions output by the suitability filter reflect the expected ground truth suitability decisions even in the presence of prediction errors as can also be seen in Figure 3. Regular recalibration and careful margin tuning ensure that $C$ continues to provide reliable estimates, even in the presence of distribution shifts or evolving deployment conditions.

## 5. Experimental Evaluation

To evaluate the performance of our proposed suitability filter, we conduct a series of experiments with different datasets,

*Table 1.* **Evaluating detection performance of the proposed suitability filter on `FMoW-WILDS`, `RxRx1-WILDS` and `CivilComments-WILDS` for $m = 0$ with both ID and OOD user data.** We report the area under the curve for ROC and PR (capturing the tradeoffs at various significance thresholds), as well as accuracy and the true false positive rate at $\alpha = 0.05$. We also report 95% confidence intervals based on 3 models $M$ trained on the same $D_{\text{train}}$ with different random seeds.

| DATASET | ACC | FPR | ROC | PR |
|---|---|---|---|---|
| `FMoW-WILDS` ID | $81.8 \pm 3.1\%$ | $0.027 \pm 0.033$ | $0.969 \pm 0.023$ | $0.967 \pm 0.029$ |
| `FMoW-WILDS` OOD | $91.9 \pm 2.5\%$ | $0.018 \pm 0.017$ | $0.965 \pm 0.016$ | $0.891 \pm 0.035$ |
| `RxRx1-WILDS` ID | $100.0 \pm 0.0\%$ | $0.000 \pm 0.000$ | $1.000 \pm 0.000$ | $1.000 \pm 0.000$ |
| `RxRx1-WILDS` OOD | $97.5 \pm 7.2\%$ | $0.031 \pm 0.088$ | $0.997 \pm 0.006$ | $0.989 \pm 0.024$ |
| `CivilComments-WILDS` ID | $93.3 \pm 5.3\%$ | $0.002 \pm 0.007$ | $0.997 \pm 0.008$ | $0.971 \pm 0.067$ |

model architectures, and naturally occurring distribution shift types from the WILDS benchmark (Koh et al., 2021).

## 5.1. General Evaluation Setup

We evaluate the suitability filter on `FMoW-WILDS` (Christie et al., 2018), `CivilComments-WILDS` (Borkan et al., 2019) and `RxRx1-WILDS` (Taylor et al., 2019). For each dataset, we follow the recommended training paradigm to train a model $M$ using empirical risk minimization and the pre-defined $D_{\text{train}} \sim \mathcal{D}_{\text{source}}$. We then further split the provided in-distribution (ID) and out-of-distribution (OOD) validation and test splits into folds as detailed in Appendix A.2.2 (16 ID and 30 OOD folds for `FMoW-WILDS`, 4 ID and 8 OOD folds for `RxRx1-WILDS`, and 16 ID folds for `CivilComments-WILDS`). We conduct two types of experiments: first, each ID fold is used as the user dataset ($D_u$), and the remaining ID data is split into 15 subsets, used as $D_{\text{test}}$ and $D_{\text{sf}}$. This yields 16×15×14 experiments for FMoW-WILDS, 4×15×14 for RxRx1-WILDS, and 16×15×14 for CivilComments-WILDS. Second, each OOD fold is used as $D_u$, and the ID data is split into 15 subsets, used for $D_{\text{test}}$ and $D_{\text{sf}}$. This yields 30×15×14 experiments for FMoW-WILDS and 8×15×14 for RxRx1-WILDS.

We define the binary suitability ground truth as $\text{Acc}(M, D_u) \geq \text{Acc}(M, D_{\text{test}}) - m$. While statistical guarantees are discussed under margin adjustments in Section 4.4, achieving the necessary calibration error estimates in practice is challenging. In particular, obtaining a reliable approximation for $\delta_{\text{target}}$ requires access to a small labeled user dataset $\hat{D}_u$, which may not always be available. Moreover, even if $\hat{D}_u$ is collected, its representativeness of the true deployment distribution $\mathcal{D}_{\text{target}}$ is uncertain, introducing potential biases in the accuracy estimation error $\Delta_u$. To account for this in our experiments, we set $m' = m$ for the non-inferiority test, effectively using the predefined margin without additional corrections. We discuss this in more detail in Appendix A.4.1. We evaluate suitability decisions by computing the ROC AUC across significance levels, capturing the trade-off between true and false posi-

tives. Additionally, we report PR AUC, accuracy, and false positive rate at the common $\alpha = 0.05$ threshold.

## 5.2. Suitability Signals

We use the following suitability signals in our instantiation of the suitability filter (more details in Appendix A.2.1):

- `conf_max`: Maximum confidence from softmax.
- `conf_std`: Standard deviation of softmax outputs, indicating confidence variability.
- `conf_entropy`: Entropy of the softmax outputs, measuring prediction uncertainty.
- `conf_ratio`: Ratio of top two class probabilities.
- `top_k_conf_sum`: Sum of the top 10% class probabilities, indicating concentration of probability mass.
- `logit_mean`: Mean of the logits, representing the overall output magnitude.
- `logit_max`: Maximum logit value, corresponding to the highest predicted class.
- `logit_std`: Standard deviation of logits, showing the spread of model outputs.
- `logit_diff_top2`: Difference between the top two logits, indicating confidence in distinguishing classes.
- `loss`: Cross-entropy loss w.r.t. the predicted class.
- `margin_loss`: Difference in cross-entropy loss between the predicted class and next best class.
- `energy`: Logits energy, computed as the negative log-sum-exponential, measuring model certainty.

## 5.3. Results

As our work introduces a novel problem setting with no existing baselines for direct comparison, the primary objective of the following is to provide an intuition for the conditions under which our approach works effectively, its limitations, and the factors influencing its performance.

Table 1 summarizes the performance of the proposed suitability filter across three benchmark datasets from the WILDS collection: `FMoW-WILDS`, `RxRx1-WILDS`, and

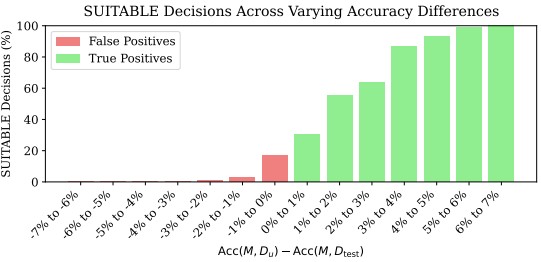

*Figure 4.* **Sensitivity of suitability decisions to accuracy differences between user and test data on `FMoW-WILDS`.** The plot, summarizing results from nearly $29k$ individual experiments, shows the percentage of SUITABLE decisions for $\alpha = 0.05$ and $m = 0$ across various accuracy difference bins. We combine both ID and OOD suitability filter experiments based on 3 models trained with different random seeds.

`CivilComments-WILDS`. Although ID and OOD results cannot be directly compared due to the differing numbers of ground truth positives and negatives, interesting trends still emerge. On `FMoW-WILDS` for example, we observe higher accuracy and a lower FPR at the $5\%$ significance level for OOD user data, while ROC AUC and PR AUC are higher for ID user data. This discrepancy may stem from class imbalance: across OOD experiments, we have nearly three times as many true negatives as true positives, making it easier to achieve high accuracy despite it generally being harder to maintain discriminative performance in an OOD setting. The latter is also confirmed for `RxRx1-WILDS`, where we see decreased performance on OOD user data compared to ID user data. Another noteworthy observation is the high overall performance on `RxRx1-WILDS`. The reason for this is that we observe large differences in model performance on `RxRx1-WILDS` depending on the fold considered, as can be seen in Table 3 (Appendix). This variation helps the suitability filter detect performance deterioration more easily, as larger performance differences enhance its ability to identify changes.

This sensitivity of suitability decisions to differences in accuracy between the user and test datasets is also illustrated in Figure 4 on `FMoW-WILDS` for $m = 0$. The ideal relationship would be a step function, where SUITABLE decisions occur only when user dataset accuracy exceeds test accuracy. However, achieving this requires a perfect estimate of accuracy on $D_u$, which is impossible without ground truth labels. In practice, we observe that the slope of the suitability decision curve is flatter than the ideal step function. There are a few erroneous SUITABLE decisions when the accuracy difference is below $0\%$, indicating occasional false positives. However, for differences $< -3\%$ (indicating a performance deterioration of at least $3\%$, this is the case for $8.4k$ experiments out of nearly $29k$ in total), our proposed suitability filter achieves $100\%$ accuracy. Additionally, some false neg-

atives are observed in the range $[0\%, 3\%]$, reflecting scenarios where the empirical evidence provided by $D_u$ and $D_{test}$ is insufficient to reject the inferiority null hypothesis at the chosen significance level $\alpha = 0.05$. However, for accuracy difference buckets exceeding $3\%$, the percentage of SUITABLE decisions consistently exceeds $80\%$ and increases to $100\%$ above $6\%$ of accuracy difference, demonstrating the robustness of the approach in scenarios with sufficiently large accuracy differences. Additional experiments, results, and interpretations can be found in Appendix A.4.

## 6. Discussion

**Conclusion.** We introduce the suitability filter, a novel framework for evaluating whether model performance on unlabeled downstream data in real-world deployment settings deteriorates compared to its performance on test data. We present an instantiation for classification accuracy that leverages statistical hypothesis testing. We provide theoretical guarantees on the false positive rate of suitability decisions and propose a margin adjustment strategy to account for calibration errors. Through extensive experiments on real-world datasets from the WILDS benchmark, we demonstrate the effectiveness of suitability filters across diverse covariate shifts. Our findings highlight the potential of suitability filters as a practical tool for model monitoring, enabling more reliable and interpretable deployment decisions in dynamic environments. Suitability filters provide an effective way to expose model capabilities and limitations and thus enable auditable service level agreements (SLAs).

**Possible Extensions.** The suitability filter framework's modularity makes it adaptable to various contexts. For fairness assessments, for instance, ensuring comparable accuracy across groups can be achieved by substituting the non-inferiority test with an equivalence test (Wellek, 2002) to evaluate if performance differences fall within a predefined margin. If the goal extends beyond snapshot evaluations to continuous monitoring, this can be achieved by applying multiple hypothesis testing corrections to the p-values. Similarly, the framework can support sequential testing, where a decision is made iteratively: a user provides an initial sample, and more data can be requested if no conclusion is reached, using methods such as O'Brien-Fleming (O'Brien & Fleming, 1979) or Pocock (Pocock, 2013) for controlling error rates. For a more detailed discussion of these extensions, we refer the interested reader to Appendix A.3.

**Limitations.** Our method is designed to detect accuracy degradations due to covariate shifts and does not address other types of distribution shift, such as label shift. This is due to the assumption that we generally only have access to unlabeled data from the target distribution $\mathcal{D}_{target}$. Future work could extend this by incorporating information from a

(potentially small) number of labeled samples from the target distribution. Moreover, our current approach is limited to classification due to the choice of signals. Though our set of suitability signals, designed to be applicable across data types, model architectures and training paradigms, provides a useful baseline, choosing signals tailored to the specific deployment setting would likely improve suitability filter performance. While our framework is general and could be used with different performance metrics, our current instantiation and experimental evaluation are limited to accuracy. It thus focuses on scenarios where average-case performance is the primary concern and does not address safety-critical applications where ensuring good performance on a per-instance (or worst-case) basis is often crucial. Lastly, it should also be noted that one of the key underlying assumptions of our framework is non-adversarial behavior from both model providers and users, who are expected to provide representative data. This assumption is justified by the user's goal of identifying a suitable model for their task, but it implies vulnerability to deliberate adversarial manipulation designed to bypass the filter.

## Code Availability

The source code for the suitability filter framework and the experiments presented in this paper is publicly available on GitHub at https://github.com/cleverhans-lab/suitability.

## Acknowledgements

We thank Anvith Thudi, Mike Menart, David Glukhov, and other members of the Cleverhans group for their feedback on this work. We would like to acknowledge our sponsors, who support our research with financial and in-kind contributions: Apple, CIFAR through the Canada CIFAR AI Chair, Meta, Microsoft, NSERC through the Discovery Grant and an Alliance Grant with ServiceNow and DRDC, the Ontario Early Researcher Award, the Schmidt Sciences foundation through the AI2050 Early Career Fellow program. Resources used in preparing this research were provided, in part, by the Province of Ontario, the Government of Canada through CIFAR, and companies sponsoring the Vector Institute.

## Impact Statement

This work proposes a suitability filter framework for evaluating machine learning models in real-world deployment settings, specifically designed to detect performance degradations caused by covariate shifts. The primary goal is to improve the robustness and fairness of machine learning models by offering tools to assess their suitability without requiring ground-truth labels. By facilitating reliable model evaluation, this work has the potential to enhance trustworthiness in automated systems especially in safety-critical deployment contexts. However, there are ethical considerations to note. The methodology assumes access to well-calibrated prediction correctness estimators, which might not hold in all scenarios, potentially leading to incorrect evaluations. Additionally, while the framework is adaptable, improper parameter choices or misinterpretations of results could exacerbate existing biases in datasets or models. Careful application and thorough understanding of the framework are critical to mitigating these risks. Future societal consequences of this work include its potential to improve fairness by enabling consistent performance evaluation across diverse subpopulations. However, misuse or overreliance on such automated evaluation frameworks without human oversight could have adverse effects. We encourage practitioners to complement this framework with domain expertise and ethical considerations during deployment. This paper aims to advance the field of Machine Learning by providing tools for model evaluation in dynamic deployment contexts. While we believe the societal implications are largely positive, we acknowledge the importance of responsibly applying this methodology to prevent unintended harm.

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

# A. Appendix

## A.1. Statistical Hypothesis Testing

### A.1.1. WELCH'S T-TEST

Welch's t-test is a modification of the standard Student's t-test that adjusts for unequal variances and unequal sample sizes between the two groups (Welch, 1947). The test statistic for the suitability filter Welch's t-test is given by:

$$t = \frac{\hat{\mu}_{D_{\text{test}}} - \hat{\mu}_{D_u}}{\sqrt{\frac{\hat{\sigma}^2_{\text{test}}}{|D_{\text{test}}|} + \frac{\hat{\sigma}^2_u}{|D_u|}}} \tag{13}$$

In the above, $\hat{\mu}_{D_u} = \frac{1}{|D_u|} \sum_{x \in D_u} p_c(x) + m$ is the margin-adjusted sample mean of $\mathbf{p_c}[D_u]$ and $\hat{\mu}_{D_{\text{test}}} = \frac{1}{|D_{\text{test}}|} \sum_{x \in D_{\text{test}}} p_c(x)$ is the sample mean of $\mathbf{p_c}[D_{\text{test}}]$. $\hat{\sigma}^2_u$ and $\hat{\sigma}^2_{\text{test}}$ are the sample variances of $\mathbf{p_c}[D_u]$ and $\mathbf{p_c}[D_{\text{test}}]$, respectively. The test statistic follows a t-distribution with degrees of freedom (df) given by:

$$\text{df} = \frac{\left( \frac{\hat{\sigma}^2_{\text{test}}}{|D_{\text{test}}|} + \frac{\hat{\sigma}^2_u}{|D_u|} \right)^2}{\frac{\left( \frac{\hat{\sigma}^2_{\text{test}}}{|D_{\text{test}}|} \right)^2}{|D_{\text{test}}|-1} + \frac{\left( \frac{\hat{\sigma}^2_u}{|D_u|} \right)^2}{|D_u|-1}} \tag{14}$$

The degrees of freedom are dependent on the size of provided user and test samples and are used to determine the appropriate critical value for the t-distribution. Since non-inferiority testing is inherently a one-sided test, after calculating the two-sample t-test statistic, the p-value is divided by 2 to reflect the one-sided nature of the non-inferiority test. This adjusted p-value is then compared to the chosen significance level $\alpha$ to determine whether the null hypothesis $H_0$ can be rejected.

### A.1.2. PROOF OF LEMMA 4.3

*Proof.* In the following, all expectations and probabilities are over samples $(x, y) \sim \mathcal{X} \times \mathcal{Y}$ unless specified otherwise. We begin by noting that the predicted correctness probability for a given sample $x$ is denoted as $p_c(x)$. We assume that $p_c(x)$ has a well-defined probability density function $f_c(\nu)$ over the interval $[0, 1]$. This means that the predicted correctness probabilities $p_c(x)$ for samples $x$ are distributed according to $f_c(\nu)$, where $\nu \in [0, 1]$ represents the possible values of prediction correctness. We can hence represent the expected value of the correctness probability $p_c(x)$ under the distribution defined by $f_c(\nu)$ as:

$$\mathbb{E}[p_c(x)] = \int_0^1 \nu f_c(\nu) \, d\nu. \tag{15}$$

The true probability of prediction correctness for a model $M$ on a sample $x$ with predicted label $M(x)$ and true label $y$ is denoted as $\mathbb{P}[M(x) = y]$. This can be expressed as the integral over all possible predicted correctness probabilities $\nu$ weighted by the conditional probability of correctness given $p_c(x) = \nu$ and the probability density $f_c(\nu)$. This decomposition follows from the law of total probability, where the predicted correctness probability $p_c(x)$ serves as an intermediate variable. We can hence write:

$$\mathbb{P}[M(x) = y] = \int_0^1 \mathbb{P}[M(x) = y \mid p_c(x) = \nu] f_c(\nu) \, d\nu. \tag{16}$$

Due to the inherent uncertainty and error in the calibration process, the predicted probability $p_c(x)$ is not necessarily equal to the true probability $\mathbb{P}_x[M(x) = y]$. We model this calibration error as $\epsilon(\nu)$, which represents the deviation of the predicted correctness from the true correctness for each possible correctness value $\nu$. Following Equation 8, we can decompose the true probability of prediction correctness as:

$$\mathbb{P}_x[M(x) = y] = \int_0^1 \nu f_c(\nu) \, d\nu + \int_0^1 \epsilon(\nu) f_c(\nu) \, d\nu = \mathbb{E}_x[p_c(x)] + \int_0^1 \epsilon(\nu) f_c(\nu) \, d\nu. \tag{17}$$

The first term represents the expected value of the predicted correctness, while the second term represents the expected error introduced by the calibration process.

Now, we make the assumption that the calibration error is equal to some small value $\delta$ with $0 \le |\delta| \ll 1$. This assumption is referred to as $\delta$-calibration as defined in Definition 4.1 (CHANGE THIS). Under this assumption, we have that:

$$\int_0^1 \epsilon(\nu) f_c(\nu) \, d\nu = \delta. \tag{18}$$

Combining this result with Equation 17, we find for the difference between the true probability and the expected correctness that:

$$\mathbb{E}[p_c(x)] - \mathbb{P}[M(x) = y] = \delta. \tag{19}$$

This completes the proof of Lemma 4.3. □

### A.1.3. PROOF OF COROLLARY 4.4

*Proof.* For simplicity, let us use the following notation:

$$\text{Acc}_{\text{source}} := \mathbb{P}_{x \sim \mathcal{D}_{\text{source}}}[M(x) = \mathcal{O}(x)], \tag{20}$$
$$\text{Acc}_{\text{target}} := \mathbb{P}_{x \sim \mathcal{D}_{\text{target}}}[M(x) = \mathcal{O}(x)], \tag{21}$$
$$\mu_{\text{source}} := \mathbb{E}_{x \sim \mathcal{D}_{\text{source}}}[p_c(x)], \tag{22}$$
$$\mu_{\text{target}} := \mathbb{E}_{x \sim \mathcal{D}_{\text{target}}}[p_c(x)]. \tag{23}$$

Here, $\mathcal{O}(x)$ represents an oracle that provides the true label $y$ for any input $x$. Let $M$ be a model with correctness estimator $C$ that is $\delta$-calibrated on both the source and target distributions. It hence follows from Lemma 4.3 that the predictions output by the correctness estimator $C$ satisfy:

$$\mu_{\text{source}} - \text{Acc}_{\text{source}} = \delta_{\text{source}} \quad \text{and} \quad \mu_{\text{target}} - \text{Acc}_{\text{target}} = \delta_{\text{target}}. \tag{24}$$

We are interested in upper-bounding the false positive rate of the end-to-end suitability filter, i.e., the probability of rejecting the null hypothesis $H_0$ at significance level $\alpha$ and returning SUITABLE when, in reality:

$$\text{Acc}_{\text{target}} < \text{Acc}_{\text{source}} - m. \tag{25}$$

We can define $m' := m + \delta_{\text{source}} - \delta_{\text{target}}$ and leverage Equation 24 to write:

$$\text{Acc}_{\text{target}} < \text{Acc}_{\text{source}} - m \iff \mu_{\text{target}} - \delta_{\text{target}} < \mu_{\text{source}} - \delta_{\text{source}} - m \iff \mu_{\text{target}} < \mu_{\text{source}} - m'. \tag{26}$$

With this margin $m'$, the corresponding null hypothesis for the non-inferiority test $H_0$ is:

$$H_0 : \mu_{\text{target}} < \mu_{\text{source}} - m'. \tag{27}$$

Assuming normalcy and independence and applying Theorem 4.2 at significance level $\alpha$, it is guaranteed that for the true mean prediction correctness $\mu_{\text{source}}$ and $\mu_{\text{target}}$ of the source and target distributions, the probability of rejecting the null hypothesis $H_0 : \mu_{\text{target}} < \mu_{\text{source}} - m'$ (i.e., concluding $\mu_{\text{target}} \geq \mu_{\text{source}} - m'$) when $H_0$ is true is controlled at $\alpha$:

$$\mathbb{P}(\text{Reject } H_0 \mid H_0 \text{ is true}) \leq \alpha. \tag{28}$$

This ensures that the difference in mean prediction correctness between the source and target distributions is bounded by the margin $m'$ with high probability.

Let us now derive the implication for the end-to-end suitability filter. By definition:

$$\begin{aligned}
\mathbb{P}(\text{Reject } H_0 \mid \text{Acc}_{\text{target}} < \text{Acc}_{\text{source}} - m) &= \frac{\mathbb{P}(\text{Reject } H_0 \cap \text{Acc}_{\text{target}} < \text{Acc}_{\text{source}} - m)}{\mathbb{P}(\text{Acc}_{\text{target}} < \text{Acc}_{\text{source}} - m)} \\
&= \frac{\mathbb{P}(\text{Reject } H_0 \cap \mu_{\text{target}} < \mu_{\text{source}} - m')}{\mathbb{P}(\mu_{\text{target}} < \mu_{\text{source}} - m')} \\
&= \mathbb{P}(\text{Reject } H_0 \mid H_0 \text{ is true}) \\
&\leq \alpha.
\end{aligned} \tag{29}$$

The inequality follows from the guarantees of the non-inferiority test as outlined in Equation 28. All other transformations are applications of Bayes' theorem and Equation 26.

Under perfect calibration, $\delta_{\text{target}} = \delta_{\text{source}} = 0$ and thus no adjustments to the performance deviation margin $m$ are needed to achieve a bounded false positive rate. Hence, under perfect calibration, the probability of rejecting the null hypothesis for a non-inferiority test with margin $m$ given that the model accuracy on $\mathcal{D}_{\text{target}}$ is lower than on $\mathcal{D}_{\text{source}}$ by more than $m$ is upper bounded by the chosen significance level $\alpha$. If we do incur miscalibration and observe $|\delta_{\text{target}}| > 0$ or $|\delta_{\text{source}}| > 0$, we have to adjust the performance deviation margin $m$ accordingly to reflect this. As shown, when choosing $m' := m + \delta_{\text{source}} - \delta_{\text{target}}$, the end-to-end suitability filter false positive rate remains bounded. This concludes the proof of Corollary 4.4.

$\square$

## A.2. Additional Experiment Details

### A.2.1. SUITABILITY SIGNALS

**General Suitability Signals.** Let $M \in \mathcal{M}$ be a classifier mapping inputs $x \in \mathcal{X}$ to probabilities over $k$ classes $\mathcal{Y} = \{1, \ldots, k\}$. Denote the logits of $M(x)$ as $z \in \mathbb{R}^k$ and the softmax outputs as $p = \text{softmax}(z)$, where $p_i = \frac{e^{z_i}}{\sum_{j=1}^{k} e^{z_j}}$. The following sample-level signals are derived from $z$ and $p$:

- Maximum confidence (`conf_max`):
$$\text{conf\_max} = \max_{i \in \{1,\ldots,k\}} p_i \tag{30}$$

  The maximum predicted probability.

- Confidence standard deviation (`conf_std`):
$$\text{conf\_std} = \sqrt{\frac{1}{k} \sum_{i=1}^{k} (p_i - \bar{p})^2}, \quad \bar{p} = \frac{1}{k} \sum_{i=1}^{k} p_i \tag{31}$$

  The standard deviation of the softmax probabilities.

- Confidence entropy (`conf_entropy`):
$$\text{conf\_entropy} = - \sum_{i=1}^{k} p_i \log(p_i + \epsilon) \tag{32}$$

  The Shannon entropy of the predicted probabilities, measuring uncertainty. We add $\epsilon = 10^{-10}$ for numerical stability.

- Confidence ratio (`conf_ratio`):
$$\text{conf\_ratio} = \frac{p_{(1)}}{p_{(2)} + \epsilon} \tag{33}$$

  The ratio of the highest to the second-highest predicted probabilities, where $p_{(1)}$ and $p_{(2)}$ are the largest and second-largest $p_i$, respectively. We add $\epsilon = 10^{-10}$ for numerical stability.

- Sum of top 10% confidences (`top_k_conf_sum`):
$$\text{top\_k\_conf\_sum} = \sum_{i \in \mathcal{K}} p_i, \quad \mathcal{K} = \text{indices of top-}\lceil 0.1k \rceil \text{ probabilities} \tag{34}$$

  The sum of the largest 10% of all predicted probabilities.

- Mean logit (`logit_mean`):
$$\text{logit\_mean} = \frac{1}{k} \sum_{i=1}^{k} z_i \tag{35}$$

  The mean of the logits.

- Maximum logit (`logit_max`):

$$\texttt{logit\_max} = \max_{i \in \{1,\dots,k\}} z_i \tag{36}$$

The maximum logit value.

- Logit standard deviation (`logit_std`):

$$\texttt{logit\_std} = \sqrt{\frac{1}{k} \sum_{i=1}^{k} (z_i - \bar{z})^2}, \quad \bar{z} = \frac{1}{k} \sum_{i=1}^{k} z_i \tag{37}$$

The standard deviation of the logits.

- Difference between two largest logits (`logit_diff_top2`):

$$\texttt{logit\_diff\_top2} = z_{(1)} - z_{(2)} \tag{38}$$

The difference between the two largest logits, where $z_{(1)}$ and $z_{(2)}$ are the largest and second-largest $z_i$, respectively.

- Loss with respect to predicted label (`loss`):

$$\texttt{loss} = -\log(p_{(1)} + \epsilon) \tag{39}$$

The cross-entropy loss with respect to the predicted label, $p_{(1)}$ is the largest $p_i$.

- Difference in loss between top two classes (`margin_loss`):

$$\texttt{margin\_loss} = -\log(p_{(1)} + \epsilon) + \log(p_{(2)} + \epsilon) \tag{40}$$

The difference in cross-entropy loss between the top two predicted probabilities. We add $\epsilon = 10^{-10}$ for numerical stability.

- Energy (`energy`):

$$\texttt{energy} = -\log \sum_{i=1}^{k} e^{z_i} \tag{41}$$

The energy function derived from the logits, measuring prediction certainty.

**Alternative Suitability Signals.** In our work, we deliberately rely on suitability signals that avoid assumptions about architecture, training, or data domains and are applicable to any classifier. However, many other signals shown to be indicative of model performance have been proposed in the literature.

In unsupervised accuracy estimation, more recent approaches measure disagreement between predictions by different models (Madani et al., 2004; Donmez et al., 2010; Platanios et al., 2016; 2017; Chen et al., 2021a; Baek et al., 2022; Jiang et al., 2021; Jaffe et al., 2015; Fan & Davidson, 2006; Yu et al., 2022; Chuang et al., 2020; Ginsberg et al., 2023), rely on manual provision of information about or make assumptions on the nature of the distribution shift between training and deployment (Redyuk et al., 2019; Chen et al., 2021b; Elsahar & Gallé, 2019; Guillory et al., 2021; Schelter et al., 2020; Peng et al., 2024; 2023; Deng & Zheng, 2021), focus on specific input data types (Maggio et al., 2022; Deng et al., 2021; Bialek et al., 2024; Sun et al., 2021; Deng & Zheng, 2021; Unterthiner et al., 2020; Guan & Yuan, 2023; Li et al., 2023), or analyze classification decision boundaries and feature separability (Hu et al., 2023; Xie et al., 2024; Tu et al., 2023; Miao et al., 2023). To ensure generality and broad applicability of the suitability filter across diverse settings, these signals are not included in our experimental evaluation. However, signals shown to predict accuracy in these studies could serve as additional suitability signals in scenarios where their specific constraints are met.

Similarly, in selective classification, recent methods enhance the underlying model by augmenting its architecture (Geifman & El-Yaniv, 2019; Lakshminarayanan et al., 2017), employing adapted loss functions during training (Gangrade et al., 2021; Huang et al., 2020; Liu et al., 2019), or utilizing more advanced prediction correctness signals, albeit often with increased inference costs (Geifman et al., 2019; Gal & Ghahramani, 2016; Rabanser et al., 2022; Feng et al., 2023). These approaches require modifications to model architecture, training processes, or inference, and are thus not generally applicable. Having said that, while these approaches are not incorporated into our work, they can serve as additional suitability signals in scenarios where these modifications are feasible.

A.2.2. DATASETS AND MODELS

**FMoW-WILDS.** The `FMoW-WILDS` dataset contains satellite images taken in different geographical regions and in different years (Christie et al., 2018; Koh et al., 2021), thus considering both temporal and geographical shift. The input $x$ is an RGB satellite image (resized to $224 \times 224$ pixels), the label $y$ is one of 62 building or land use categories, and the domain represents the year the image was taken and its geographical region. We train a DenseNet-121 model (Huang et al., 2017) pretrained on ImageNet (Russakovsky et al., 2015) and without $L_2$ regularization with empirical risk minimization. We use the Adam optimizer (Kingma, 2014) with an initial learning rate of $10^{-4}$ that decays by 0.96 per epoch, and train fro 50 epochs with early stopping and a batch size of 64. All reported results are averaged over 3 random seeds. Following the standard WILDS training setup, we use $76,863$ images from the years 2002-2013 as training data. We split the remaining ID and OOD splits into 16 different ID folds and 30 different OOD data folds as detailed in Table 2. These folds were chosen with the aim to be as representative as possible of shifts likely to occur in practice while still ensuring a sufficient number of samples per fold for statistical testing (at least 666).

**RxRx1-WILDS.** The `RxRx1-WILDS` dataset reflects the disribution shifts induced by batch effects in the context of genetic perturbation classification (Taylor et al., 2019; Koh et al., 2021). The input $x$ is a 3-channel image of human cells obtained by fluorescent microscopy (nuclei, endoplasmic reticuli and actin), the label $y$ indicates which of the 1,139 genetic treatments (including no treatment) the cells received, and the domain specifies the batch in which the imaging experiment was run. The images in `RxRx1-WILDS` are the result of executing the same experiment 51 times, each in a different batch of experiments. Each experiment was run in a single cell type, one of: HUVEC (24 experiments), RPE (11 experiments), HepG2 (11 experiments), and U2OS (5 experiments) across 2 sites. The dataset is split by experimental batches into training, validation, and test sets. For all experiments, we fine-tune a ResNet-50 model (He et al., 2016) pretrained on ImageNet (Russakovsky et al., 2015), using a learning rate of $10^{-4}$ and L2-regularization strength of $10^{-5}$. The models are trained with the Adam optimizer (Kingma, 2014) and a batch size of 75 for 90 epochs, linearly increasing the learning rate for 10 epochs and then decreasing it following a cosine learning rate schedule. Results are reported averaged over 3 random seeds. Following the standard WILDS training setup, we use $40,612$ images from 33 experiments in site 1 as training data. We then split the remaining data by cell type into 4 ID data folds (same experiments as training data but different images but site 2) and 8 OOD data folds (different experiments, combining sites 1 and 2) as detailed in Table 3.

**CivilComments-WILDS.** The `CivilComments-WILDS` dataset focuses on text toxicity classification across demographic identities, aiming to address biases in toxicity classifiers that can spuriously associate toxicity with certain demographic mentions (Borkan et al., 2019; Koh et al., 2021). The input $x$ is a text comment on an online article, and the label $y$ is whether the comment was rated as toxic or not. The domain is represented as an 8-dimensional binary vector, where each component corresponds to the mention of one of the 8 demographic identities: male, female, LGBTQ, Christian, Muslim, other religions, Black, and White. The dataset consists of 450,000 comments, annotated for toxicity and demographic mentions by multiple crowdworkers and randomly split into train, validation and test splits. We hence have no additional OOD data splits (and correspondingly, no OOD data folds) for this dataset. We train a DistilBERT-base-uncased model (Sanh, 2019) with the AdamW optimizer (Loshchilov et al., 2017), using a learning rate of $10^{-5}$, a batch size of 16, and an L2 regularization strength of $10^{-2}$ for 5 epochs with early stopping. All reported results are averaged over 3 random seeds. Following the standard WILDS training setup, we use $269,038$ comments as training data. We split the remaining data into 16 different ID folds as detailed in Table 4. Since the data is generally from the same distribution as our training data but we divide it into folds depending on the sensitive attributes mentioned in each comment, this is an example of the target distribution consisting of subpopulations of the source distribution.

**A.3. Possible Extensions**

A.3.1. EQUIVALENCE TESTING

In equivalence testing, the goal is to assess whether the performance on the target dataset $\mathcal{D}_{\text{target}}$ is statistically similar to the performance on the source dataset $\mathcal{D}_{\text{source}}$ within a specified margin $m$, i.e., we want to test whether the difference between the two means is sufficiently small (Wellek, 2002). This is formalized as the following hypothesis setup:

$$H_0 : |\mu_{\text{target}} - \mu_{\text{source}}| > m \tag{42}$$

$$H_1 : |\mu_{\text{target}} - \mu_{\text{source}}| \leq m \tag{43}$$

*Table 2.* Summary of different ID and OOD data folds for `FMoW-WILDS`. For accuracy, we report the mean and the 95% confidence interval based on three models trained with different random seeds.

| SPLIT | YEAR | REGION | NUM SAMPLES | ACCURACY |
|---|---|---|---|---|
| | | ID FOLDS | | |
| ID_VAL | 2002-2006 | ALL | 1420 | $53.99 \pm 1.32\%$ |
| ID_VAL | 2007-2009 | ALL | 1430 | $55.78 \pm 0.70\%$ |
| ID_VAL | 2010 | ALL | 2459 | $62.60 \pm 1.64\%$ |
| ID_VAL | 2011 | ALL | 2874 | $65.98 \pm 1.25\%$ |
| ID_VAL | 2012 | ALL | 3300 | $64.03 \pm 0.13\%$ |
| ID_VAL | ALL | ASIA | 2693 | $62.42 \pm 1.03\%$ |
| ID_VAL | ALL | EUROPE | 5268 | $59.88 \pm 0.56\%$ |
| ID_VAL | ALL | AMERICAS | 3076 | $63.85 \pm 2.10\%$ |
| ID_TEST | 2002-2006 | ALL | 1473 | $51.39 \pm 4.54\%$ |
| ID_TEST | 2007-2009 | ALL | 1423 | $57.25 \pm 0.61\%$ |
| ID_TEST | 2010 | ALL | 2456 | $61.01 \pm 0.41\%$ |
| ID_TEST | 2011 | ALL | 2837 | $65.03 \pm 1.15\%$ |
| ID_TEST | 2012 | ALL | 3138 | $62.42 \pm 1.36\%$ |
| ID_TEST | ALL | ASIA | 2615 | $59.39 \pm 1.94\%$ |
| ID_TEST | ALL | EUROPE | 5150 | $58.99 \pm 0.51\%$ |
| ID_TEST | ALL | AMERICAS | 3130 | $63.05 \pm 1.39\%$ |
| | | OOD FOLDS | | |
| VAL | 2013 | ALL | 3850 | $60.29 \pm 1.81\%$ |
| VAL | 2014 | ALL | 6192 | $62.44 \pm 1.48\%$ |
| VAL | 2015 | ALL | 9873 | $57.77 \pm 1.37\%$ |
| VAL | ALL | ASIA | 4121 | $56.30 \pm 0.73\%$ |
| VAL | ALL | EUROPE | 7732 | $63.28 \pm 1.07\%$ |
| VAL | ALL | AFRICA | 803 | $50.73 \pm 1.25\%$ |
| VAL | ALL | AMERICAS | 6562 | $58.04 \pm 2.05\%$ |
| VAL | ALL | OCEANIA | 693 | $66.38 \pm 2.51\%$ |
| VAL | 2013 | EUROPE | 1620 | $61.30 \pm 1.11\%$ |
| VAL | 2014 | EUROPE | 2523 | $68.05 \pm 1.94\%$ |
| VAL | 2015 | EUROPE | 3589 | $60.82 \pm 0.73\%$ |
| VAL | 2013 | ASIA | 813 | $57.40 \pm 3.89\%$ |
| VAL | 2014 | ASIA | 1311 | $56.90 \pm 2.31\%$ |
| VAL | 2015 | ASIA | 1997 | $55.45 \pm 0.26\%$ |
| VAL | 2013 | AMERICAS | 1168 | $61.13 \pm 1.66\%$ |
| VAL | 2014 | AMERICAS | 1967 | $60.85 \pm 1.26\%$ |
| VAL | 2015 | AMERICAS | 3427 | $55.36 \pm 3.32\%$ |
| TEST | 2016 | ALL | 15959 | $55.48 \pm 1.14\%$ |
| TEST | 2017 | ALL | 6149 | $48.64 \pm 2.13\%$ |
| TEST | ALL | ASIA | 4963 | $55.67 \pm 0.72\%$ |
| TEST | ALL | EUROPE | 5858 | $56.38 \pm 1.96\%$ |
| TEST | ALL | AFRICA | 2593 | $33.50 \pm 3.87\%$ |
| TEST | ALL | AMERICAS | 8024 | $56.20 \pm 1.17\%$ |
| TEST | ALL | OCEANIA | 666 | $59.56 \pm 0.43\%$ |
| TEST | 2016 | EUROPE | 4845 | $58.42 \pm 2.68\%$ |
| TEST | 2017 | EUROPE | 1013 | $46.63 \pm 1.48\%$ |
| TEST | 2016 | ASIA | 3216 | $53.58 \pm 0.80\%$ |
| TEST | 2017 | ASIA | 1747 | $59.53 \pm 1.42\%$ |
| TEST | 2016 | AMERICAS | 6165 | $57.21 \pm 1.42\%$ |
| TEST | 2017 | AMERICAS | 1859 | $52.86 \pm 1.49\%$ |

The null hypothesis $H_0$ asserts that the difference in means between the target and source distributions is greater than the margin $m$. The alternative hypothesis $H_1$ posits that the means are equivalent, with their difference being smaller than or equal to the margin $m$. In practice, this is achieved by conducting two one-sided tests (TOST). This involves testing both lower and upper bounds of the margin to confirm that the performance difference is not meaningfully large in either direction.

*Table 3.* Summary of different ID and OOD data folds for `RxRx1-WILDS`. For accuracy, we report the mean and the 95% confidence interval based on three models trained with different random seeds.

| SPLIT | CELL TYPE | NUM SAMPLES | ACCURACY |
|---|---|---|---|
| | | ID FOLDS | |
| ID_TEST | HEPG2 | 8622 | $25.39 \pm 1.43\%$ |
| ID_TEST | HUVEC | 19671 | $50.30 \pm 1.24\%$ |
| ID_TEST | RPE | 8623 | $23.86 \pm 1.17\%$ |
| ID_TEST | U2OS | 3696 | $17.00 \pm 0.98\%$ |
| | | OOD FOLDS | |
| VAL | HEPG2 | 2462 | $21.01 \pm 1.77\%$ |
| VAL | HUVEC | 2464 | $36.85 \pm 0.10\%$ |
| VAL | RPE | 2464 | $16.44 \pm 0.96\%$ |
| VAL | U2OS | 2464 | $2.27 \pm 0.30\%$ |
| TEST | HEPG2 | 7388 | $22.63 \pm 1.28\%$ |
| TEST | HUVEC | 17244 | $39.99 \pm 1.13\%$ |
| TEST | RPE | 7360 | $21.32 \pm 0.41\%$ |
| TEST | U2OS | 2440 | $8.96 \pm 1.78\%$ |

*Table 4.* Summary of different ID data folds for `CivilComments-WILDS`. For accuracy, we report the mean and the 95% confidence interval based on three models trained with different random seeds.

| SPLIT | SENSITIVE ATTRIBUTE | NUM SAMPLES | ACCURACY |
|---|---|---|---|
| VAL | MALE | 4765 | $89.31 \pm 0.22\%$ |
| VAL | FEMALE | 5891 | $90.09 \pm 0.68\%$ |
| VAL | LGBTQ | 1457 | $80.00 \pm 0.97\%$ |
| VAL | CHRISTIAN | 4550 | $92.72 \pm 0.17\%$ |
| VAL | MUSLIM | 2110 | $81.52 \pm 1.39\%$ |
| VAL | OTHER RELIGIONS | 986 | $85.87 \pm 0.77\%$ |
| VAL | BLACK | 1652 | $77.85 \pm 1.14\%$ |
| VAL | WHITE | 2867 | $77.26 \pm 0.76\%$ |
| TEST | MALE | 14295 | $88.84 \pm 0.16\%$ |
| TEST | FEMALE | 16449 | $90.02 \pm 0.18\%$ |
| TEST | LGBTQ | 4426 | $79.78 \pm 0.68\%$ |
| TEST | CHRISTIAN | 13361 | $92.22 \pm 0.25\%$ |
| TEST | MUSLIM | 6982 | $82.65 \pm 0.58\%$ |
| TEST | OTHER RELIGIONS | 3500 | $88.18 \pm 0.58\%$ |
| TEST | BLACK | 4872 | $78.39 \pm 0.68\%$ |
| TEST | WHITE | 7969 | $79.88 \pm 0.47\%$ |

### A.3.2. CONTINUOUS MONITORING

In continuous monitoring, the aim is to regularly re-evaluate if a model is still suitable for a given deployment context based on new incoming data samples. When performance is evaluated over time with changing data, the Benjamini-Hochberg (BH) procedure is used to control the false discovery rate (FDR) across multiple tests (Benjamini & Hochberg, 1995). The BH procedure adjusts p-values by considering the number of tests performed up to the current point, ensuring that the proportion of false positives remains controlled. This is formalized as follows: for each p-value $p_i$, the null hypothesis is rejected if $p_i \leq \frac{i}{m} \cdot \alpha$, where $m$ is the total number of tests and $\alpha$ is the desired FDR threshold. The rolling window approach further refines this by evaluating significance across a fixed window of recent data, smoothing out short-term fluctuations and focusing on long-term trends in performance. This approach helps identify true changes in model performance while accounting for variations in individual datasets over time.

### A.3.3. SEQUENTIAL TESTING

When testing the same hypothesis sequentially with accumulating data, the O'Brien-Fleming (O'Brien & Fleming, 1979) and Pocock (Pocock, 2013) methods are used to control the overall false positive rate (type I error rate) across multiple tests.

These methods are designed for sequential testing, where a decision is made at each stage based on the data collected so far, and more data can be added if no conclusion is reached. The O'Brien-Fleming method is more conservative early on, requiring stronger evidence to reject the null hypothesis at earlier stages and relaxing this criterion as more data becomes available. Specifically, the significance threshold at stage $k$ is adjusted as:

$$\alpha_k = 1 - (1 - \alpha)^{1/(n-k+1)} \tag{44}$$

where $n$ is the total number of stages and $\alpha$ is the desired overall Type I error rate. In contrast, the Pocock method applies a constant critical value across all stages of testing. For each stage $k$, the significance level remains:

$$\alpha_k = \frac{\alpha}{n} \tag{45}$$

Both methods adjust the significance threshold at each stage to control the family-wise error rate (FWER), ensuring that the probability of making at least one Type I error remains below a specified threshold, $\alpha$.

### A.4. Additional Results

#### A.4.1. CALIBRATION

**Impact of Calibration on $\mathcal{D}_{\text{target}}$.** We visualize the impact of calibration on classifier $C$ and the performance of the end-to-end suitability filter in Figure 5. To this end, we select the ID validation and test splits from Europe as $D_{\text{sf}}$ and $D_{\text{test}}$, respectively. We then plot the actual accuracy versus the mean of the prediction correctness estimated by a classifier $C$ trained on $D_{\text{sf}}$, with or without additional calibration on $D_u$. It should be noted that this is mainly a theoretical experiment, as in practice calibration on $D_u$ is not possible since we do not have access to ground truth information for user data. We observe that the false positive rate of the end-to-end suitability filter is elevated due to miscalibration on the different distribution $\mathcal{D}_{\text{target}}$. Although the relationship between actual accuracy and mean estimated prediction correctness is weaker without calibration, these metrics remain highly correlated. Therefore, the increased risk from miscalibration can be mitigated by selecting an appropriate non-inferiority margin $m$.

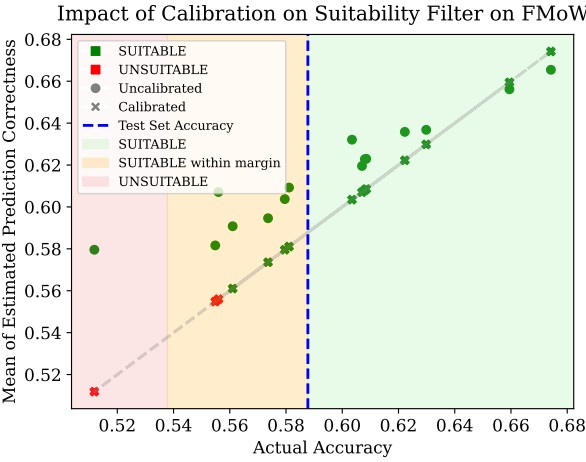

*Figure 5.* Suitability filtering on different OOD folds of `FMoW-WILDS` with and without additional calibration on $D_u$. We choose a non-inferiority margin of $m = 0.05$ for this experiment.

**Accuracy Estimation Error.** In Section 4.4, we propose using the empirical accuracy estimation error $\Delta$ to adjust the margin and mitigate the effects of miscalibration in $C$. To illustrate this in practice, Figure 6 presents the distribution of $\Delta$ for both test and user data across 6300 experiments on `FMoW-WILDS`. As expected, $\Delta_{\text{test}}$ is centered around zero, indicating that the estimated accuracy closely matches the ground truth accuracy and there is no clear directional bias. However, $\Delta_u$ is frequently positive, indicating that accuracy is often overestimated. This miscalibration can lead to incorrect suitability decisions. While adjusting the performance deterioration margin $m$, as proposed in Section 4.4, would mitigate this issue,

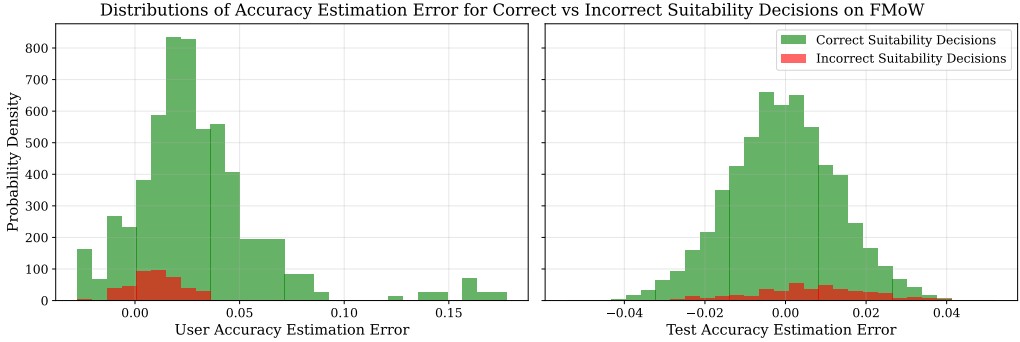

*Figure 6.* Distribution of the empirical accuracy estimation error $\Delta$ for both the user and the test data across 6300 experiments on `FMoW-WILDS`. The suitability decisions depicted here have been made for a choice of $m = 0$ without margin adjustment due to miscalibration and at a significance level of $\alpha = 0.05$.

no such adjustment was applied here to highlight the impact of miscalibration on suitability decisions. Notably, suitability decision errors do not occur for examples with large accuracy estimation errors.

To better understand this phenomenon, Figure 7 explores the relationship between accuracy estimation error $\Delta_u$ and the actual performance degradation from test to user data. As performance deteriorates, the accuracy estimation error tends to increase. However, performance degradation grows at a faster rate than $\Delta_u$, meaning that the overall impact of $\Delta_u$ on suitability decisions remains limited for $m = 0$. This explains why incorrect suitability decisions are primarily concentrated near the decision boundary rather than in cases with extreme accuracy estimation errors (and, correspondingly, larger performance degradation).

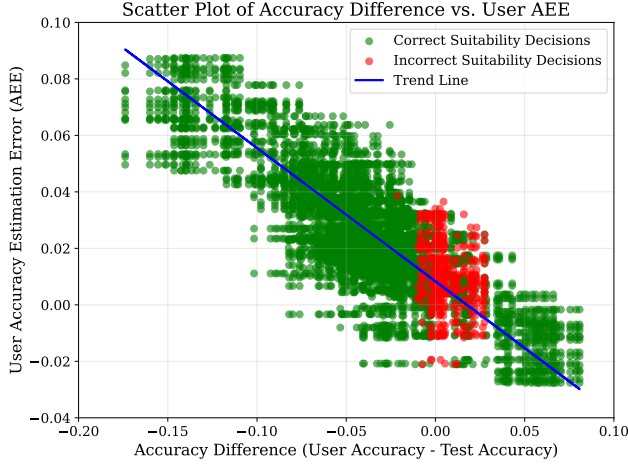

*Figure 7.* Relationship between performance deterioration for model $M$ and the empirical accuracy estimation error $\Delta$ for the user data across 6300 experiments on `FMoW-WILDS`. The suitability decisions depicted here have been made for a choice of $m = 0$ without margin adjustment due to miscalibration and at a significance level of $\alpha = 0.05$. As can be seen, incorrect suitability decisions are centered around the suitability decision boundary and are not, as might be expected, in areas of large empirical accuracy estimation error $\Delta$.

By incorporating margin adjustments based on the empirical accuracy estimation error, suitability decisions can be made more robust against calibration errors, ultimately improving the reliability of the suitability filter in deployment.

A.4.2. USING DIFFERENT SIGNAL SUBSETS FOR PREDICTION CORRECTNESS ESTIMATOR

In Table 5, we compare our proposed suitability filter that trains the prediction correctness estimator using various suitability signals to alternatives that rely on only a single signal. As can be seen, the suitability filter leveraging all signals

generally outperforms single-signal alternatives, demonstrating the benefit of integrating diverse signals for robust suitability decisions. However, we find that certain signals, such as `energy` or `logit_max`, perform nearly as well on their own. Unsurprisingly, these signals are also identified as the most predictive of per-sample prediction correctness for classifier $C$ (see Appendix A.4.3). Noteworthy outliers in Table 5 include `logit_mean` and `logit_std` that have relatively high accuracy but higher FPR and lower ROC and PR AUC than comparable signals. Upon closer examination, we find that prediction correctness classifiers $C$ trained on only these signals generally have a higher expected calibration error even when tested on in-distribution data as can be seen in Table 6. As demonstrated in Corollary 4.5, proper calibration is theoretically crucial for reliable suitability decisions, and this importance is evident in practice here. Signals that yield the best-performing prediction correctness estimators $C$ (high accuracy and low maximum calibration error) also demonstrate superior performance when applied in the end-to-end suitability filter.

*Table 5.* Comparing performance of the proposed suitability filter against individual signal-based suitability decisions on `FMoW-WILDS` for $m = 0$ with both ID and OOD user data. We report the area under the curve for ROC and PR (capturing the tradeoffs at various significance thresholds), as well as accuracy and the true false positive rate at $\alpha = 0.05$. We also report 95% confidence intervals based on 3 models $M$ trained on the same $D_{\text{train}}$ with different random seeds.

| METHOD | ACC | FPR | ROC | PR |
|---|---|---|---|---|
| | | ID USER DATA | | |
| SUITABILITY FILTER | $81.8 \pm 3.1\%$ | $0.027 \pm 0.033$ | $0.969 \pm 0.023$ | $0.967 \pm 0.029$ |
| ENERGY | $80.6 \pm 4.3\%$ | $0.024 \pm 0.040$ | $0.965 \pm 0.020$ | $0.962 \pm 0.033$ |
| LOGIT_MAX | $80.2 \pm 4.8\%$ | $0.025 \pm 0.041$ | $0.965 \pm 0.018$ | $0.963 \pm 0.030$ |
| LOGIT_MEAN | $80.1 \pm 10.4\%$ | $0.112 \pm 0.194$ | $0.918 \pm 0.113$ | $0.896 \pm 0.196$ |
| LOGIT_DIFF_TOP2 | $73.5 \pm 4.5\%$ | $0.008 \pm 0.001$ | $0.963 \pm 0.017$ | $0.963 \pm 0.017$ |
| MARGIN_LOSS | $73.5 \pm 4.5\%$ | $0.008 \pm 0.001$ | $0.963 \pm 0.017$ | $0.963 \pm 0.017$ |
| LOGIT_STD | $72.3 \pm 13.3\%$ | $0.170 \pm 0.134$ | $0.855 \pm 0.144$ | $0.779 \pm 0.300$ |
| CONF_ENTROPY | $71.1 \pm 2.4\%$ | $0.003 \pm 0.012$ | $0.969 \pm 0.014$ | $0.967 \pm 0.012$ |
| CONF_STD | $68.8 \pm 3.1\%$ | $0.008 \pm 0.019$ | $0.963 \pm 0.020$ | $0.960 \pm 0.017$ |
| TOP_K_CONF_SUM | $68.2 \pm 6.9\%$ | $0.005 \pm 0.015$ | $0.947 \pm 0.027$ | $0.944 \pm 0.029$ |
| CONF_MAX | $67.9 \pm 4.4\%$ | $0.008 \pm 0.021$ | $0.957 \pm 0.025$ | $0.954 \pm 0.025$ |
| LOSS | $67.0 \pm 2.8\%$ | $0.008 \pm 0.021$ | $0.952 \pm 0.031$ | $0.948 \pm 0.035$ |
| CONF_RATIO | $62.3 \pm 4.5\%$ | $0.046 \pm 0.053$ | $0.846 \pm 0.056$ | $0.826 \pm 0.016$ |
| | | OOD USER DATA | | |
| SUITABILITY FILTER | $91.9 \pm 2.5\%$ | $0.018 \pm 0.017$ | $0.965 \pm 0.016$ | $0.891 \pm 0.035$ |
| ENERGY | $91.9 \pm 4.7\%$ | $0.008 \pm 0.007$ | $0.971 \pm 0.005$ | $0.910 \pm 0.028$ |
| LOGIT_MAX | $91.9 \pm 4.7\%$ | $0.008 \pm 0.007$ | $0.971 \pm 0.005$ | $0.910 \pm 0.030$ |
| CONF_ENTROPY | $89.1 \pm 3.1\%$ | $0.011 \pm 0.020$ | $0.957 \pm 0.007$ | $0.872 \pm 0.078$ |
| CONF_STD | $88.9 \pm 4.0\%$ | $0.010 \pm 0.019$ | $0.952 \pm 0.013$ | $0.854 \pm 0.108$ |
| LOGIT_DIFF_TOP2 | $88.9 \pm 2.7\%$ | $0.005 \pm 0.011$ | $0.976 \pm 0.014$ | $0.917 \pm 0.074$ |
| MARGIN_LOSS | $88.9 \pm 2.7\%$ | $0.005 \pm 0.011$ | $0.976 \pm 0.014$ | $0.917 \pm 0.074$ |
| CONF_MAX | $88.4 \pm 4.2\%$ | $0.011 \pm 0.020$ | $0.948 \pm 0.015$ | $0.842 \pm 0.121$ |
| LOSS | $88.3 \pm 3.4\%$ | $0.012 \pm 0.023$ | $0.944 \pm 0.014$ | $0.831 \pm 0.118$ |
| TOP_K_CONF_SUM | $86.7 \pm 4.4\%$ | $0.026 \pm 0.057$ | $0.916 \pm 0.005$ | $0.773 \pm 0.097$ |
| CONF_RATIO | $83.6 \pm 6.4\%$ | $0.001 \pm 0.005$ | $0.905 \pm 0.050$ | $0.711 \pm 0.102$ |
| LOGIT_MEAN | $61.7 \pm 20.5\%$ | $0.446 \pm 0.268$ | $0.845 \pm 0.193$ | $0.698 \pm 0.175$ |
| LOGIT_STD | $28.3 \pm 16.3\%$ | $0.812 \pm 0.166$ | $0.324 \pm 0.256$ | $0.137 \pm 0.019$ |

### A.4.3. SIGNAL IMPORTANCE FOR THE PREDICTION CORRECTNESS ESTIMATOR

To analyze the importance of individual signals used to estimate prediction correctness, we present the ANOVA results on `FMoW-WILDS` in Table 7. All signals, except for `class_prob_ratio`, show extremely high F-values with corresponding p-values essentially zero, indicating their strong statistical significance in explaining the variance in prediction correctness. The most valuable signals, as indicated by the highest F-values in Table 7, are `logit_max`, `energy`, `margin_loss`, and `logit_diff_top2`. For certain signals, the sign of the logistic regression coefficient matches our expectations, with a higher `logit_max` value, an increase in the logit difference for the predicted class and the runner-up (`logit_diff_top2`) or low `energy` indicating a correct prediction. Interestingly, however, we also observe that for features such as `conf_max`, the sign is negative, indicating that lower confidence is indicative of higher likelihood of correct prediction. While this seems counterintuitive at first, it should be noted that for a large majority of samples this signal is 1 and is hence heavily

*Table 6.* Table showcasing the mean accuracy and calibration metrics (ECE, MCE, RMSCE) for prediction correctness estimators trained on different signals, with 95% confidence intervals. Suitability Filter refers to the classifier $C$ trained using all available signals. The metrics evaluate the classifiers' prediction quality and their calibration over 3 random splits of the FMoW-WILDS ID train and validation data splits.

| SIGNALS | ACCURACY | ECE | MCE | RMSCE |
|---|---|---|---|---|
| SUITABILITY FILTER | $77.5 \pm 0.3\%$ | $0.021 \pm 0.007$ | $0.055 \pm 0.021$ | $0.027 \pm 0.006$ |
| LOGIT_MAX | $76.6 \pm 0.9\%$ | $0.027 \pm 0.017$ | $0.068 \pm 0.050$ | $0.033 \pm 0.020$ |
| ENERGY | $76.3 \pm 0.9\%$ | $0.029 \pm 0.012$ | $0.075 \pm 0.054$ | $0.035 \pm 0.017$ |
| MARGIN_LOSS | $75.5 \pm 0.6\%$ | $0.021 \pm 0.003$ | $0.060 \pm 0.021$ | $0.028 \pm 0.002$ |
| LOGIT_DIFF_TOP2 | $75.5 \pm 0.6\%$ | $0.021 \pm 0.003$ | $0.060 \pm 0.021$ | $0.028 \pm 0.002$ |
| CONF_ENTROPY | $74.4 \pm 0.9\%$ | $0.089 \pm 0.007$ | $0.208 \pm 0.055$ | $0.113 \pm 0.009$ |
| CONF_STD | $73.2 \pm 0.6\%$ | $0.104 \pm 0.020$ | $0.263 \pm 0.068$ | $0.134 \pm 0.013$ |
| CONF_MAX | $72.7 \pm 0.5\%$ | $0.116 \pm 0.014$ | $0.260 \pm 0.056$ | $0.143 \pm 0.015$ |
| LOSS | $72.0 \pm 0.4\%$ | $0.121 \pm 0.012$ | $0.265 \pm 0.036$ | $0.148 \pm 0.022$ |
| TOP_K_CONF_SUM | $67.7 \pm 1.8\%$ | $0.146 \pm 0.026$ | $0.301 \pm 0.066$ | $0.177 \pm 0.030$ |
| LOGIT_MEAN | $67.1 \pm 2.3\%$ | $0.081 \pm 0.033$ | $0.268 \pm 0.081$ | $0.119 \pm 0.046$ |
| CONF_RATIO | $61.1 \pm 2.9\%$ | $0.144 \pm 0.008$ | $0.320 \pm 0.008$ | $0.187 \pm 0.001$ |
| LOGIT_STD | $59.8 \pm 2.4\%$ | $0.132 \pm 0.070$ | $0.294 \pm 0.209$ | $0.167 \pm 0.108$ |

*Table 7.* ANOVA results showing the significance of individual signals in predicting model correctness. Signals are ordered by decreasing F-value, which measures the variance explained by each signal relative to the residual variance. We also include the sign of the model's coefficients for each signal, indicating whether a given feature positively or negatively influences the prediction correctness estimate.

| SIGNAL | F-VALUE | P-VALUE | REL. |
|---|---|---|---|
| LOGIT_MAX | 2090.61 | 0 | + |
| ENERGY | 2051.45 | 0 | − |
| MARGIN_LOSS | 1982.44 | 0 | − |
| LOGIT_DIFF_TOP2 | 1978.44 | 0 | + |
| CONF_ENTROPY | 1390.30 | 0 | − |
| CONF_STD | 1232.76 | 0 | − |
| CONF_MAX | 1108.32 | 0 | − |
| LOSS | 934.26 | 0 | − |
| LOGIT_MEAN | 755.29 | 0 | − |
| TOP_K_CONF_SUM | 281.76 | 0 | + |
| LOGIT_STD | 116.16 | $9.18 \cdot 10^{-27}$ | − |
| CONF_RATIO | 12.05 | $5.21 \cdot 10^{-4}$ | + |

concentrated around 0 after normalization. The contribution from this signal is thus mostly relevant in cases where the maximum confidence is below 1 anyways, in which case it seems that a higher confidence can be indicative of incorrect predictions.

SHAP (SHapley Additive exPlanations) is a model-agnostic method for interpreting machine learning models by assigning each feature a contribution value to the model's prediction. It calculates Shapley values based on cooperative game theory, ensuring that the contribution of each feature is fairly distributed by considering all possible feature combinations and their impact on the prediction. As can be seen in Figure 8, the signals deemed most predictive of prediction correctness are the same ones as identified by the ANOVA analysis in Table 7.

### A.4.4. CHOICE OF MODEL ARCHITECTURE FOR PREDICTION CORRECTNESS ESTIMATOR

In Table 8, we compare the accuracy of different prediction correctness estimators on the FMoW-WILDS dataset. We evaluate a range of classifiers, including simple models like Logistic Regression and more complex architectures such as Single-Layer and Two-Layer Neural Networks. We observe that logistic regression performs as well as more complex models, delivering high accuracy and low expected calibration error. While this may seem surprising, it is important to note that the suitability signals are already non-linear transformations of the model's output logits. Since these transformations capture the key relationships needed for our task, using more complex models capable of learning additional non-linear patterns, such as neural networks, does not provide any further benefit.

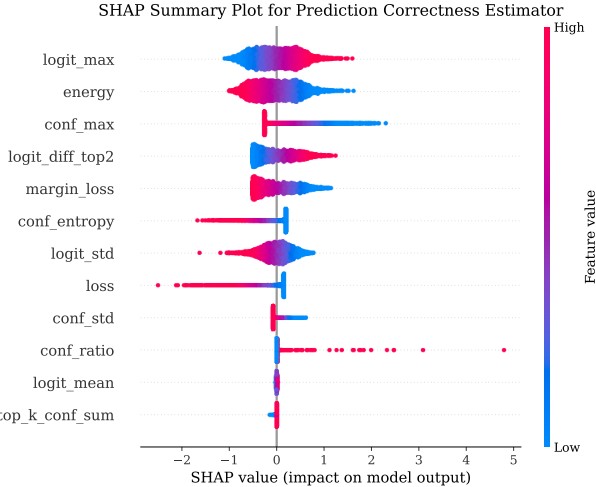

*Figure 8.* SHAP analysis for the prediction correctness estimator on `FMoW-WILDS`.

*Table 8.* Table showcasing the mean accuracy and calibration metrics (ECE, MCE, RMSCE) for various classifiers, with $95\%$ confidence intervals. The metrics evaluate the classifiers' prediction quality and their calibration over 3 random splits of the `FMoW-WILDS` ID train and validation splits.

| CLASSIFIER | ACCURACY | ECE | MCE | RMSCE |
|---|---|---|---|---|
| LOGISTIC REGRESSION | $77.2 \pm 1.4\%$ | $0.022 \pm 0.004$ | $0.055 \pm 0.040$ | $0.027 \pm 0.007$ |
| GRADIENT BOOSTING CLASSIFIER | $77.1 \pm 1.3\%$ | $0.020 \pm 0.018$ | $0.062 \pm 0.102$ | $0.027 \pm 0.030$ |
| SINGLE-LAYER NEURAL NETWORK | $77.1 \pm 1.2\%$ | $0.037 \pm 0.006$ | $0.086 \pm 0.017$ | $0.046 \pm 0.006$ |
| SUPPORT VECTOR MACHINE | $77.1 \pm 0.9\%$ | $0.077 \pm 0.034$ | $0.232 \pm 0.147$ | $0.107 \pm 0.050$ |
| RANDOM FOREST | $76.5 \pm 0.9\%$ | $0.031 \pm 0.007$ | $0.067 \pm 0.068$ | $0.037 \pm 0.016$ |
| TWO-LAYER NEURAL NETWORK | $75.7 \pm 2.3\%$ | $0.040 \pm 0.006$ | $0.087 \pm 0.012$ | $0.048 \pm 0.002$ |
| DECISION TREE | $70.7 \pm 0.9\%$ | $0.111 \pm 0.021$ | $0.161 \pm 0.050$ | $0.123 \pm 0.029$ |

