# OpenReview forum: "Suitability Filter: A Statistical Framework for Classifier Evaluation in Real-World Deployment Settings"
_ICML.cc/2025/Conference — ICML 2025 oral_

### Official Review · Reviewer_kVMa · 2025-03-04

**Overall Recommendation:** 5

**Summary:**

This paper presents a new problem of predicting whether the model's performance on a runtime dataset (called user dataset) has decreased compared to the test dataset. To tackle this problem, they introduce a method called suitability filter, which consists in two main steps: 1. several sample-level performance scores from the literature are computed and fused together with a learned regression model. The output of the regression model should predict correctness of individual samples. 2.  A statistical test is conducted on fused sample signals to predict whether the whole user dataset differs in performance from the test dataset.

The authors present several theoretical proofs regarding the efficiency of their approach under restrictive assumptions.

They conduct experiments to test their approach on benchmark datasets from WILDS.

**Claims And Evidence:**

In view of the experimental results, the method seems effective in practice.

My doubt is regarding the applicability/usefulness of the theoretical results.
How are these bounds used in practice? Are they actually verified from the experiments? I did not find answers to these questions in the manuscript.

**Essential References Not Discussed:**

Nothing specific

**Experimental Designs Or Analyses:**

The experiments are sufficient and well designed.
However, I believe that Section 5.1 is really hard to follow. Experimental description should be made more simple. Why introducing new variables (n, m, k) to present the experimental setup? It would be much clearer to simply replace these values with the true values that were actually used in the experiments.

**Methods And Evaluation Criteria:**

The proposed evaluation is sufficient and rigorous.
They do not compare with existing literature but it is OK as the problem presented is new.

However, I would have liked to see an ablation study to see the importance of the two contributions.
For instance, I would have liked to see how their signal fusion approach works to detect errors on individual samples.
As this is a widely studied problem, this would have allowed for comparison against the literature.

**Other Comments Or Suggestions:**

- This work focuses only on classification tasks. This should be reflected in the title: Suitability Filter: A Statistical Framework for Classification Model Evaluation in Real-World Deployment Settings.

- Definition 3.2: The use of both "if and only if" and "with high probability" in the same sentence seem contradicting.

- Why the name Dsf? What does it stand for?

- In eq 2 and 3, same notations (x1) represent different things.

- I am not sure to understand the meaningfulness of Theorem 4.2 and its practical relevance and the insights that it brings. Is the assumption of independant and normally distributed samples really relevant? Where is it used later on? If it is only used in the proofs, maybe it should be placed in the appendix.

- Typo in caption of table 1: =rate

- Two different definitions associated with parameter alpha: In the experiments it is the FP rate whereas in the theroem 4.2 it is different.

**Other Strengths And Weaknesses:**

The ideas presented in this paper are very interesting.
Overall the presented framework is generic and modular, which is good.

However, I sometimes felt like the notations and concepts presentations were overly complex, even for simple ideas, making the paper hard to follow.

Regarding the introduction, I think it would have been worth discussing further the relevance and applicability domain of the newly introduced problem. Unlike most existing methods, they propose to not consider data suitability at the sample level, but rather at the dataset level, by aggregating acceptance/rejection signals. This is an interesting take, which is interesting for some applications but not all. For example, the credit risk model presented in the introduction is a good use case where dataset level suitability is interesting. However, other cases such as autonomous systems might not respond very well to such kind of safety monitors. It would be good to give more examples and to delineate applicability scenarios in the introduction.

**Questions For Authors:**

See questions and comments from other sections.

**Relation To Broader Scientific Literature:**

The literature in the broad topic of runtime monitoring of neural networks is becoming very broad, but the literature review proposed in the paper is sufficient to introduce the new problem and present the novelty of the work.

**Theoretical Claims:**

I did not check the details of the proofs.

---

> ### Author Rebuttal · Authors · 2025-04-01
>
> We thank the reviewer for their time spent assessing our paper and for their critical and detailed feedback. We particularly appreciate their positive evaluation of our work’s novelty, the framework’s modularity and our experimental evaluation. We provide our answers to the reviewer’s raised concerns below, are looking forward to their response, and hope for a favorable reassessment of our score.
>
> **C1: Practical applicability of theoretical results.**
> As can be seen in Figure 5, margin adjustments become increasingly important the further away we move from the test set accuracy. On FMoW-WILDS, adjusting a 5% margin with e.g., 500 labeled user samples reduces the false positive rate from unbounded to below 0.03, satisfying a 0.05 significance level. The number of labelled samples available plays a crucial role here: the higher the number of samples, the more accurate the estimates of $\Delta_u$ and thus, the more precise the margin adjustment that leads to theoretical guarantees.
>
> **C2: Importance of the prediction correctness estimator for the end-to-end suitability filter performance.**
> Our prediction correctness estimator leverages multiple suitability signals—several of which have been independently proposed in the selective classification literature (see Section 2)—yielding performance on par with or better than any single signal for misclassification prediction. The contribution of the statistical test to the end-to-end suitability filter performance is orthogonal to this, as it allows for the aggregation of prediction correctness estimates across data samples.
>
>
> **C3: Clarity of Experimental Setup (Section 5.1).**
> We thank the reviewer for remarking that Section 5.1 is hard to follow and propose to rewrite the section as follows:
> "We evaluate the suitability filter on FMoW-WILDS, CivilComments-WILDS and RxRx1-WILDS. For each dataset, we follow the recommended training paradigm to train a model using empirical risk minimization and the pre-defined $D_{\text{train}}$. We then further split the provided in-distribution (ID) and out-of-distribution (OOD) validation and test splits into folds as detailed in Appendix A.2.2 (16 ID and 30 OOD folds for FMoW-WILDS, 4 ID and 8 OOD folds for RxRx1-WILDS, and 16 ID folds for CivilComments-WILDS). We conduct two types of experiments: first, each ID fold is used as the user dataset ($D_u$​), and the remaining ID data is split into 15 subsets, used as $D_\text{test}$ and $D_\text{sf}$​. This yields 16×15×14 experiments for FMoW-WILDS, 4×15×14 for RxRx1-WILDS, and 16×15×14 for CivilComments-WILDS. Second, each OOD fold is used as $D_u$, and the ID data is split into 15 subsets, used for $D_\text{test}$ and $D_\text{sf}$​​. This yields 30×15×14 experiments for FMoW-WILDS and 8×15×14 for RxRx1-WILDS."
>
>
> **C4: Delineation of applicability scenarios.**
> We thank the reviewer for highlighting the potential limitations of our suitability monitors in autonomous systems. We agree that for critical applications, focusing on average-case suitability is not sufficient. Instead, ensuring good performance on a per-instance (or worst-case) basis is crucial, a problem setting that is addressed in the selective classification literature. Our work, however, focuses on scenarios where average-case suitability is the primary concern. We will ensure to clarify this in future versions of the paper.
>
>
> **C5: Possibly contradictory statement in Definition 3.2.**
> We respectfully disagree with the concern that using “if and only if” and “with high probability” in the same sentence is contradictory. As per Definitions 3.1 and 3.2, "if and only if" defines the deterministic rule for outputting “SUITABLE,” while "with high probability" captures the uncertainty in estimating model performance (due to sampling or variance). Once the high-probability condition is met—i.e., the model's performance on $D_{u}$ is within a margin $m$ of its performance on $D_\text{test}$—the filter deterministically outputs “SUITABLE.”
>
>
> **C6: Complexity of notations and nomenclature.**
> We tried to strike a balance between theoretical rigor and avoiding overly complex notations. Thus, in both equations 2 and 3, $x_1$ represents the first sample of the respective dataset. $D_\text{sf}$ stands for the **s**uitability **f**ilter data that is used to learn the prediction correctness estimator.
>
>
> **C7: Meaningfulness of Theorem 4.2.**
> This theorem, drawn from established results (Lehmann et al., 1986; Wellek, 2002), underpins our theoretical analysis. The assumption of independent, normally distributed samples is standard for many tests, allowing us to use these guarantees. Here, $\alpha$ denotes the significance level, the probability of a false positive.
>
>
> Lastly, we will consider changing the title to reflect that our work focuses on classification only as suggested by the reviewer. We hope that we have addressed the reviewer’s concerns and are looking forward to their response.

---

### Official Review · Reviewer_RLS2 · 2025-03-14

**Overall Recommendation:** 4

**Summary:**

This paper proposes a new paradigm for detecting whether the performance on unlabeled data processed during deployment, falls by a certain margin below the performance on a held-out dataset sampled from the training distribution but used for evaluation. This is a novel framework for detecting performance deterioration in deployment settings under covariate shift.

**Claims And Evidence:**

The results indicate that the proposed approach shows good performance on both in-distribution and out-of-distribution experiments across different tasks and datasets.

**Essential References Not Discussed:**

The related works section describes different orthogonal ways to do different parts of their approach, and why and how they differ from the proposed approach.

**Experimental Designs Or Analyses:**

The paper uses feature groups in the benchmark datasets to understand out-of-distribution performance. This approach is sound as these groups are naturally occurring. The suitability signals seem to cover different metrics informative to the suitability filer.

**Methods And Evaluation Criteria:**

This paper evaluates on different tasks and corresponding benchmark datasets. The datasets for classification span land use classification, text toxicity classification, and genetic perturbation classification.

**Other Comments Or Suggestions:**

N/A

**Other Strengths And Weaknesses:**

N/A

**Questions For Authors:**

N/A

**Relation To Broader Scientific Literature:**

Given the rise in deployed machine-learning models in safety-critical systems, this work is highly relevant.

**Theoretical Claims:**

I have looked at the proofs at a high level, but they are beyond my area of expertise. I think that it would be best if someone with more expertise could kindly check the details.

---

> ### Author Rebuttal · Authors · 2025-03-31
>
> We thank the reviewer for their time spent assessing our paper, and appreciate their positive evaluation of our work’s relevance, novelty, and soundness.

---

### Official Review · Reviewer_i1Vd · 2025-03-16

**Overall Recommendation:** 4

**Summary:**

The authors developed a framework to evaluate how well a trained model will perform when deployment for real-world inference.
The framework combines ideas from distribution shift detection, selective inference, and interestingly dataset inference.
A specific instantiation of the framework is presented and evaluated on datasets from the WILDS benchmark collection.

**Claims And Evidence:**

One claim is unwarranted: "we are able to detect performance deterioration of more than 3% with 100% accuracy".
Also the claim about being agonistic to the model architecture needs to be substantiated.
Otherwise, the claims are reasonable.

**Essential References Not Discussed:**

It would quite helpful to relate the presented solution with the statical view of uncertainty by Gruber et al:
"Sources of Uncertainty in Supervised Machine Learning – A Statisticians’ View"
https://arxiv.org/abs/2305.16703

**Experimental Designs Or Analyses:**

Reasonable. WILDS is a a great benchmark of in-the-wild distribution shifts.

**Methods And Evaluation Criteria:**

Yes

**Other Comments Or Suggestions:**

I encountered a few typos:
- degredation => degradation
- are no limited => not
- deplyoment

**Other Strengths And Weaknesses:**

This is a neat work that builds on the progress in distribution shift detection to provide a comprehensive and practical solution to make decision on the suitability of an ML model at deployment time.

Some highlights:
+ careful usage of statistical measures to instantiate the proposed framework and calibrate it against a target dataset.
+ statical guarantees backed by theoretical analysis.
+ the authors discuss several practical aspects such as continuous monitoring.

Some recommendations:
- Provide more details about the model architectures used in the evaluation. Demonstrate quantiatively that you approach is agnostic to the architecture used.
- Provide examples that the framework do not predict well. It is not hard to introduce delibrate perturbations to a dataset that degrades the performance significantly while passing all the checks you introduce.
- Consider breaking down uncertainty into aleatoric and epistemic as proposed by Gruber et al  (see reference cited above).
- Provide figures to illustrate some key concepts such as calibration.

**Questions For Authors:**

Can the developed framework be instantiated for evaluating adversarial robustness?
Similarly, I would curious to see if adversarially-trained models fare better in handling distribution shifts as measured by your framework.

**Relation To Broader Scientific Literature:**

The presented work combines ideas from closely related topics and utilizes them to design a practical framework.

**Theoretical Claims:**

I skimmed over the proofs of Lemma 4.3. and COROLLARY 4.4. There are a few assumptions introduced but they are reasonable.

---

> ### Author Rebuttal · Authors · 2025-04-01
>
> We thank the reviewer for their time spent assessing our paper and for their critical and detailed feedback. We particularly appreciate their positive evaluation of our work’s practical applicability, theoretical foundation and experimental design. We provide our answers to the reviewer’s raised concerns below, are looking forward to their response, and hope for a favorable reassessment of our score.
>
>
> **C1: Claim of detecting performance deterioration of more than 3% with 100% accuracy.**
> We assume that the reviewer refers to key contribution 4 at the end of the introduction section. It should be clarified that this is not a general performance guarantee but an empirical finding that has been validated across 29k experiments on FMoW-WILDS. We have made sure to clearly mention this in an updated revision of the paper and to point interested readers to Figure 4, which visualizes this finding.
>
>
> **C2: Claim of model architecture agnosticism.**
> The proposed suitability filter framework is inherently compatible with various model architectures because it only requires the model to be a classifier, without making any other assumptions. As mentioned in the introduction, we propose a well-performing default instantiation of the filter. This instantiation uses domain-agnostic suitability signals and can be applied to any classifier that outputs logits, regardless of its underlying architecture. To show this empirically, we evaluate this instantiation on diverse architectures including a DenseNet-121, a ResNet-50 and a DistilBERT-base-uncased model. Additional details about our experiments are provided in Appendix A.2.2. We hope that this addresses the reviewer’s concern and welcome any further questions.
>
>
> **C3: Connection to Gruber et al.**
> We appreciate the reviewer’s suggestion to consider the decomposition of uncertainty into aleatoric and epistemic factors, as proposed by Gruber et al. However, this approach does not directly align with our primary objectives. Our method does not rely on quantifying or separating out specific sources of uncertainty; rather, it compares predictive performance deviations between the user’s dataset and the model provider’s test set. This viewpoint effectively marginalizes over varying error sources that might lead to a decrease in accuracy, including aleatoric and epistemic uncertainty. Nonetheless, we acknowledge that such a decomposition may be valuable as an alternate instantiation of suitability filters beyond accuracy (i.e., determining suitability based on deviations in uncertainty beyond a margin $m$).
>
>
> **C4: Adversarial examples and the limitations of suitability filters.**
> We thank the reviewer for their insightful comments regarding deliberate perturbations and adversarial robustness. There seem to be different angles to this concern that we would like to address separately:
>
> 1. Deliberate perturbation of examples: The reviewer correctly pointed out that a model user could deliberately perturb their data to be misclassified by the model without these perturbations being detected by the suitability filter. One of the key underlying assumptions of our framework is that both the model provider and the model user are non-adversarial and provide representative samples of their datasets. We believe this is realistic because the model user's goal is to identify the most suitable model for their actual task. Deliberately perturbing their own data samples to pass a suitability check would undermine this goal. We will clarify this assumption in the paper.
> 2. Using suitability filters to evaluate adversarial robustness: While we have not confirmed this experimentally, we do think it should be possible to use suitability filters to evaluate adversarial robustness. We will leave it to future work to explore suitability filters where suitability is defined based on other metrics than accuracy.
> 3. Effects of distribution shift for adversarially-trained models: Our framework could indeed be used to evaluate whether adversarially trained models exhibit greater robustness to distribution shifts. Since the suitability filter is independent of the training algorithm, it could be used to compare the suitability of two models on the same user data, one of which was trained normally and the other with adversarial training.
>
>
> **C5: Figures to illustrate calibration.**
> We visualize the effect of miscalibration and motivate the margin adjustment under accuracy estimation error in Figure 3. In Figure 5 in the Appendix, we empirically show the effect of miscalibration on the FMoW-WILDS dataset. We would be happy to include additional figures to illustrate calibration if the reviewer has a specific figure in mind.
>
>
> Lastly, we thank the reviewer for pointing out the typos and we will make sure to correct them in future versions of our paper. We hope that we have addressed the reviewer’s concerns and are looking forward to their response.

---

> > ### Comment · Reviewer_i1Vd · 2025-04-05
> >
> > I appreciate the detailed response of the authors. I stay with my rating.

---

### Decision · Program_Chairs · 2025-05-01

**Decision:**

Accept (oral)

**Comment:**

The proposed suitability filter framework effectively addresses the crucial challenge of detecting model performance deterioration on unlabeled user data compared to labeled test data. All reviewers acknowledge the work's novelty, theoretical soundness, and practical utility for safety-critical applications. My recommendation is to accept the paper.